# Cohort profile: The ENTWINE iCohort study, a multinational longitudinal web-based study of informal care

Saif Elayan[1]*, Eva Bei[2], Giulia Ferraris[3], Oliver Fisher[4], Mikołaj Zarzycki[5], Viola Angelini[1], Lena Ansmann[6,7], Erik Buskens[8], Mariët Hagedoorn[3], Milena von Kutzleben[6], Giovanni Lamura[9], Anne Looijmans[3], Robbert Sanderman[3], Noa Vilchinsky[2], Val Morrison[10]

1 Faculty of Economics and Business, Department of Economics, Econometrics and Finance, University of Groningen, Groningen, The Netherlands, 2 Faculty of Social Sciences, Department of Psychology, Bar-Ilan University, Ramat Gan, Israel, 3 Department of Health Psychology, University Medical Center Groningen, University of Groningen, Groningen, The Netherlands, 4 Department of Economics and Social Sciences, Università Politecnica delle Marche, Ancona, Italy, 5 Department of Psychology, Liverpool Hope University, Liverpool, United Kingdom, 6 Department of Health Services Research, Division of Organizational Health Services Research, University of Oldenburg, Oldenburg, Germany, 7 Faculty of Medicine, Institute of Medical Sociology, Health Services Research and Rehabilitation Science (IMVR), University of Cologne, Cologne, Germany, 8 Department of Epidemiology, University Medical Center Groningen, University of Groningen, Groningen, The Netherlands, 9 IRCCS INRCA-National Institute of Health and Science on Ageing, Centre for Socio-Economic Research on Ageing, Ancona, Italy, 10 School of Human and Behavioural Sciences, Bangor University, Bangor, United Kingdom

* s.y.i.elayan@rug.nl

**Data Availability Statement:** A de-identified minimal data set underlying the results described in this article is available on the Zenodo repository at https://doi.org/10.5281/zenodo.8170318.

## Abstract

Informal care is a key pillar of long-term care provision across Europe and will likely play an even greater role in the future. Thus, research that enhances our understanding of caregiving experiences becomes increasingly relevant. The ENTWINE iCohort Study examines the personal, psychological, social, economic, and geographic factors that shape caregiving experiences. Here, we present the baseline cohort of the study and describe its design, recruitment methods, data collection procedures, measures, and early baseline findings. The study was conducted in nine countries: Germany, Greece, Ireland, Israel, Italy, the Netherlands, Poland, Sweden, and the United Kingdom. The study comprised a web-based longitudinal survey (baseline + 6-month follow-up) and optional weekly diary assessments conducted separately with caregivers and care recipients. From 14 August 2020 to 31 August 2021, 1872 caregivers and 402 care recipients were enrolled at baseline. Participants were recruited via Facebook and, to a lesser extent, via the study website or caregiver/patient organisations. Caregiver participants were predominantly female (87%) and primary caregivers (82%), with a median age of 55 years. A large proportion (80%) held at least post-secondary education, and two-thirds were married/partnered. Over half of the caregivers were employed (53%) and caring for a person with multiple chronic conditions (56%), and nearly three-quarters were caring for either a parent (42%) or a spouse/partner (32%). About three-quarters of care recipient participants were female (77%), not employed (74%), and had at least post-secondary education (77%), with a median age of 55 years. Over half of the care recipients were married/partnered (59%), receiving care primarily from

Additional baseline data requests can be made after September 2024 to the ENTWINE data access committee via email at entwine@umcg.nl.

**Funding:** This research project is funded by the European Union's Horizon 2020 research and innovation programme under the Marie Skłodowska-Curie Innovative Training Network (H2020-MSCA-ITN-2018), grant agreement No. 814072. The funders had no role in study design, data collection and analysis, decision to publish, or preparation of the manuscript.

**Competing interests:** The authors have declared that no competing interests exist.

their spouses/partners (61%), and diagnosed with multiple chronic conditions (57%). This study examining numerous potential influences on caregiving experiences provides an opportunity to better understand the multidimensional nature of these experiences. Such data could have implications for developing caregiving services and policies, and for future informal care research.

## Introduction

The population of Europe is shrinking and growing older, albeit to differing degrees and at different rates in different countries [1]. Increases in life expectancy and reduced fertility rates are the main drivers of this transition [2]. Life expectancy at birth has been steadily increasing in Europe since the 1960s at a rate of more than two years per decade, reaching an average of 81 years in 2018 [3,4]. Also, over the same period, the average life expectancy at age 65 years in Europe rose by more than five years, with 65-year-olds in 2018 expected to live around 18.1 to 21.6 more years [4–6]. As the prevalence of disabilities, chronic diseases and frailty increases with advancing age, an unprecedented increase in the number of people in need of long-term care has been noted in the past few decades [7–9]. This trend is expected to continue in the next 50 years, with average life expectancy at birth reaching 86.1 and 90.3 years for men and women, respectively, by 2070 [10]. Although European countries have responded to this emerging demand in different ways, the dominant trend is unmistakable: policy reforms that increasingly shift the responsibility for long-term care from the welfare state to informal caregivers [11,12]. As a result, greater expectations and responsibility are now being placed on informal caregivers, who are expected to play an even greater role than their already dominant one in the provision of long-term care. The prevalence of informal care in Europe is already large, with estimates ranging from 9% to 34% among the adult population [13]. However, recent research projects that the number of potential caregivers available per older person will decline uniformly in the upcoming decades, thus increasing pressure on fewer available caregivers [14]. Considering the growing role of informal caregivers, it is imperative that we seek ways to ameliorate the potential negative effects of informal care on their health, wellness, and financial well-being so that the informal caregiving 'system' is sustained. Therefore, research that seeks to understand caregiving experiences and outcomes, as well as the underlying factors that shape them, is becoming increasingly relevant and necessary.

As noted in the literature [15,16], informal care is a complex phenomenon, where multiple factors interplay to shape its provision as well as the associated experience and outcomes. These factors include psychological (e.g., personality, attitudes and motivations, relational affection), contextual (e.g., time availability, distance, and the health status of the care recipient) and social (e.g., social network size, perceived support) factors that can either facilitate or hinder care provision, and either exacerbate or ameliorate associated negative experiences and outcomes [17–28]. Informal care decisions and experiences can also be shaped by several macro-level factors, including the availability of support services for informal caregivers, community and private services (e.g., migrant care), as well as assistive technologies [29–31]. Furthermore, although informal care is often treated as a 'free' resource in long-term care systems, it is now well established that informal caregivers' time might not be costless, while they also incur substantial economic [32,33] and non-economic (e.g., health and well-being) costs [34,35]. Also, these costs are disproportionate among different groups of informal caregivers (e.g., caregivers of patients with different types of conditions) and are associated with negative caregiving outcomes [35–39].

Given the complex nature of the informal caregiving experience, it is crucial that any attempt to comprehend it takes into account the various intertwined factors that contribute to shaping this experience. With this view in mind, we conducted the ENTWINE iCohort Study on informal care. The ENTWINE iCohort Study is a multi-purpose web-based study with an intensive longitudinal design established in 2020 to examine the range of personal, interpersonal, psychological, social, economic, and geographic factors that may co-shape caregiving experiences and outcomes for diverse groups of informal caregivers and care recipients, and for society, in nine different countries that have different care systems. As such, it offers a unique opportunity for a more nuanced understanding of the caregiving experience and hopefully contributes to the introduction or extension of caregiver-centred support.

The study is conducted by the ENTWINE consortium, a Marie Skłodowska-Curie Innovation Training Network (ITN) funded by the European Commission through the Horizon 2020 research programme. The consortium brings together senior and early-stage researchers to investigate a broad spectrum of challenges in informal caregiving and issues concerning the development and use of innovative psychology-based and technology-based interventions that support the willingness and opportunity to provide informal care.

The objective of this paper is to present the baseline cohort of the ENTWINE iCohort Study and describe the cohort design, recruitment, data collection procedures, measures, early baseline findings, and data access procedure.

## Materials and methods

### Design and recruitment of collaborating countries

The ENTWINE iCohort Study is a multinational web-based cohort study with an intensive longitudinal design that combines a two-wave panel survey (baseline + 6 months follow-up) with optional weekly diary assessments. The entire data collection period spanned from August 2020 to December 2021. The cohort includes caregivers and care recipients from nine countries, including those represented in the ENTWINE consortium (the United Kingdom, the Netherlands, Italy, Sweden, and Israel) and four other European countries (Germany, Greece, Poland, and Ireland). Participating countries were selected to represent different geographic areas (i.e., North, East, West, and South) and typologies of welfare states in Europe [40]. The initial plan was to administer the surveys in paper and web-based formats; however, the former format was suspended due to COVID-19 pandemic restrictions. The detailed protocol of the study has been published elsewhere [41].

### Study setting and population

The ENTWINE iCohort Study was conducted in eight European countries and Israel. The total population of participating countries is approximately 300 million, of which around 244 million are aged 18 years or older [42,43]. According to data from the European Quality of Life Survey 2016, the prevalence of informal care in participating European countries amongst adults aged 18 or older ranged from 10% in Ireland to 34% in Greece [13]. In Israel, 30% of adults aged 20 or older are informal caregivers [44]. Thus, the estimated total number of adult informal caregivers in all participating countries is approximately 50 million.

The study inclusion criteria, assessed by means of an eligibility survey preceding the baseline survey, were: 1) residing in one of the participating countries; 2) being able to answer the surveys in English, Swedish, German, Dutch, Italian, Greek, Hebrew, or Polish; 3) having access and being able to use the Internet; 4) being aged 18 years or above; and 5) having the self-declared cognitive and physical capacity to complete the surveys. In addition, informal caregivers had to be providing care to an adult (aged ≥ 18 years) with a chronic health condition, disability or any

other care need. For care recipients, they had to be receiving care from an adult (aged $\geq$ 18 years) as a result of a chronic health condition, disability, or any other care need.

## Cohort recruitment

To recruit participants, we employed a coordinated recruitment strategy encompassing digital and non-digital methods. Face-to-face recruitment was also initially planned but cancelled due to COVID-19 lockdowns. Non-digital recruitment methods included radio and newspaper advertisements, conference announcements, distribution of flyers in care settings, and word-of-mouth referrals. Digital recruitment methods included social media postings, and calls for participation published in participating universities' newsletters. Facebook was selected as the primary social media platform for recruitment due to its popularity, advanced targeting and reporting options, as well as the familiarity of the research team with it. Our Facebook recruitment efforts began with free posts on the dedicated Facebook pages we designed for the study (in the eight languages offered), as well as on Facebook pages or groups related to caregivers, older adults, and patients with chronic diseases. To take advantage of Facebook's ability to target specific audiences, we initiated paid targeted advertisements from December 2020 to May 2021. This included image and video advertisements we made visible to Facebook users who met the inclusion criteria regarding age and country of residence. In addition, we further targeted potential participants through the use of keywords and interest groups. Facebook's performance metrics were regularly monitored to gauge and optimise recruitment performance. Free posts about the study were also published by members of the consortium on Twitter, Instagram and LinkedIn, albeit to a lesser extent than on Facebook.

To further enhance recruitment, we also approached local and international caregiver organisations (e.g., Carers Trust, UK), as well as advocacy groups for older adults and patients with chronic diseases (e.g., The Brain Charity, UK) in order to ask them to distribute emails or newsletters inviting participation in the study to their members. Organisations and groups were not offered any incentives for recruitment efforts. A complete list of all caregiver organisations and advocacy groups involved in the recruitment of participants is shown in S1 Table. We also used snowball sampling by asking participants if they would agree to provide the email address of their caregiver or care recipient in order to invite them to participate in the study. Inviting the caregiver or care recipient was optional, and individual participation in the study was not contingent upon it.

Irrespective of the recruitment method, all flyers, emails, newsletters and social media postings included information about the study and its eligibility criteria and were available multilingually as appropriate. Recruitment materials also included a (hyper)link to the study website (https://www.entwine-icohort.eu), a contact email address for queries and comments, as well as a (hyper)link or/and a scanning QR code directing to the eligibility survey. Participants were not offered any form of compensation for participating in the study.

## Data collection procedure

Access to the surveys was provided via Questback Enterprise Feedback Suite® [45], a specialised survey platform for online data collection. Surveys were accessible via a computer, laptop or any other smart device (e.g., smartphones and tablets). Participants were able to exit any of the survey(s) and return later to complete it from where they left off—even on another device. Cookies were not used in order to maintain confidentiality and to allow multiple respondents to complete the surveys using the same device (e.g., a device shared by a caregiver and their care recipient). In accordance with the General Data Protection Regulation (GDPR) 2016/679, respondents' IP addresses were not recorded or made available to the research team.

As a first step, potential participants who were directed to the link to the eligibility survey were required to answer a series of screening questions to confirm their eligibility before enrolment. Those confirmed as eligible were then required to read a participant information sheet and a consent form and to provide their e-mail address. Upon providing their consent and email address, each participant was sent an invitation email. This email included copies of the participant information sheet and the signed digital consent form, as well as a personalised link to the baseline web-based survey. The survey link was valid for three weeks and, if necessary, a maximum of two reminder emails were automatically sent within a week of the invitation. All participants invited to participate in the baseline survey after 31 May 2021 were informed that they would not take part in the follow-up study as they could not meet the deadline for data collection (15 December 2021).

On accessing the baseline survey link, participants were automatically assigned unique pseudonyms (i.e., artificial identifiers) by Questback to obscure their email addresses. This pseudonymisation technique was used to prevent duplicate participation and protect participants' rights to privacy while still allowing for longitudinal data collection and matching responses across study surveys. After being invited to complete the baseline survey, participants, regardless of whether they completed all parts of the baseline survey, were invited to participate in a weekly diary study. The weekly diary assessments were delivered online once per week for 24 consecutive weeks. Participants in the diary study received various reminder and motivational emails. At the end of the diary study period, regardless of whether they had completed all weeks or not, participants were invited to the follow-up survey. The invitations were sent via email and contained a personalised link to the follow-up survey that was valid for two weeks. As in the baseline survey, reminder emails were sent to those who had not completed the follow-up survey, three and seven days after receiving the invitation.

## Surveys

**Survey environment.**   All study surveys were built using Questback Enterprise Feedback Suite® [45]. In addition to pre-made Questback library questions and themes, free programming languages including HyperText Markup Language (HTML), Cascading Style Sheets (CSS), the Hypertext Preprocessor (PHP), and JavaScript were used to create working personalised dynamic surveys. Dynamic features included question routing (i.e., hiding questions based on answers or combinations of answers to previous questions), text substitution (i.e., feeding answers into subsequent questions), database links (e.g., presenting the surveys in participant's indicated preferred language), randomisation of survey modules (see below), and response validation (e.g., pointing out missing questions, incorrect response format, and potential data entry mistakes through plausibility checks). These dynamic features contributed to the user-friendliness of the survey, low data entry errors, as well as favourable response and completion rates.

**Surveys' contents.**   The eligibility survey started with a question about the language (amongst all offered languages) in which potential participants would prefer to complete the survey. Respondents were also asked how they had heard about the study in order to inform ongoing recruitment efforts. Respondents were next presented with a brief eligibility screener based on the predefined inclusion and exclusion criteria described earlier: this included three questions on age, country of residence, and their caregiving or care-receiving status. The latter question was a single-answer multiple-choice question with the following answer options: "I provide care for a family member or a friend with a chronic health condition, disability or any other care need that is 18 years old or over"; "I receive care from a family member or a friend, that is 18 years old or older, due to my chronic health condition, disability or any other care

need"; and "None of the above". Based on their answer to this question, eligible participants were assigned to either the caregiver or care recipient sample. Conversely, respondents who selected "None of the above" were deemed ineligible, as they did not meet the criteria of being caregivers or care recipients, and thus, were screened out and not included in the study. We purposefully used a wide definition of informal care without restrictions on the health condition(s) and the level of impairment of the care recipient; the social relationship between the caregiver and care recipient; or the intensity of caregiving. The last two questions concerned whether participants would like to be contacted about future research and whether they would like to receive a summary of key findings upon study completion.

Caregiver and Care Recipient baseline surveys comprised five module sections; a core module and four additional modules, each addressing a specific theme related to informal caregiving. The core module assessed sociodemographics and aspects of the care situation and included validated questionnaires targeting key dimensions of the caregiving experience (e.g., well-being, willingness to provide care, and relationship characteristics). The additional four modules were dedicated to specific themes: Module 1, cultural and psychosocial aspects; Module 2, personality and geographical barriers; Module 3, interpersonal processes; and Module 4, employment, costs and use of formal care services (including migrant care).

In the Caregiver Baseline Survey, all caregivers were asked to complete the core module (approximately 20 to 30 minutes to complete) and three randomly assigned and ordered modules (each approximately 10 minutes to complete). The decision to randomly assign three out of the four additional modules was made to shorten the survey and reduce its response burden, thereby improving the response rate. The Care Recipient Baseline Survey was shorter (approximately 40 minutes to complete); therefore, care recipients were asked to complete the core module and all four additional modules (randomly ordered).

Caregiver and Care Recipient Follow-up surveys included the same questions as the baseline surveys (minus demographics) as well as additional questions about whether participants still provide/receive care. A summary of the measures used in the ENTWINE iCohort baseline surveys is provided in Table 1. Details about the measures used for this study are reported in a previous paper [41].

In this article, we focus on characterising the cohort at baseline in terms of key sociodemographics (i.e., age, gender, marital status, education and employment status), caregiving situation characteristics (i.e., care intensity, primary caregiving, and kinship type), and care recipient's health condition(s) and dependency. Furthermore, we consider the willingness and ability to provide care, gains and burden from caregiving, and well-being.

The dependency level of care recipients was assessed using the Katz Index of Independence in Activities of Daily Living (ADL). This instrument measures independence in performing six basic ADL: bathing, dressing, toileting, transferring, continence, and feeding. Each of these ADL is scored as 1 for independence and 0 for dependence, and the Katz index score is obtained by totalling these individual scores [47]. The score indicates complete independence (score = 6), partial dependence (score 3–5), or severe dependence (score ≤ 2) [73].

Caregiver willingness and ability to provide care were measured using the Willingness to Care Scale. The scale comprises 30 items, each representing a specific emotional, instrumental, or nursing care task. The ability to perform each task was scored as either "able" or "not able", and the willingness to carry out the task was rated on a 5-point Likert scale, ranging from "completely unwilling" (1) to "completely willing" (5). The ability-to-care score was computed by summing up the "able" responses, while the willingness-to-care score was calculated by averaging the ratings on the 5-point Likert scale [51].

Caregiver gains were assessed using the GAINS scale. The scale comprises ten items, each scored on a 4-point Likert scale ranging from 0 ("not at all") to 3 ("a lot"). The total score was

**Table 1. A summary of the measures used in the ENTWINE iCohort baseline surveys.**

| Module/Topic | Measure(s) | Max number of items | Answer type | CG[a] Baseline Survey | CR[b] Baseline Survey |
|---|---|---|---|---|---|
| **Core Module** | | | | | |
| Caregiver sociodemographic characteristics | Items concerning age; gender; home location; the highest level of attained education; partnership status; the number of dependants, siblings and living parents; employment status; income; religion; ethnicity; migration background; the number of care recipients; own health condition(s); relationship to the care recipient; caregiving duration and frequency; the presence of other caregiver(s); previous care experience; and distance to the care recipient | 47 | Yes/no, multiple-choice, and numerical/ text fields | ✓ | ✗ |
| Care recipient sociodemographic characteristics (as reported by the caregiver) | Items concerning age, gender, health condition(s) of the care recipient, length of illness, and living arrangements | 32 | Yes/no, multiple-choice, and numerical/ text fields | ✓ | ✗ |
| Care recipient sociodemographic characteristics | Items concerning age; gender; home location; the highest level of attained education; partnership status; the number of dependants, siblings and living parents; employment status; income; religion; ethnicity; migration background; own health condition(s); relationship to the caregiver; caregiving duration and frequency; the presence of other caregiver(s); living arrangements; and distance to the caregiver | 58 | Yes/no, multiple-choice, and numerical/ text fields | ✗ | ✓ |
| The importance of religion | Single item: What is the importance of religion in your life? | 1 | Likert scale | ✓ | ✓ |
| Perceived interpersonal connectedness | Inclusion of Other in the Self Scale (IOS) [46] | 1 | Likert scale | ✓ | ✓ |
| The capacity of the care recipient to perform activities of daily living | Katz Index of Independence in Activities of Daily Living (ADL) [47] | 6 | Yes/no | ✓ | ✓ |
| Time devoted to helping with household activities, personal care, and practical and emotional support | Caregiver Indirect and Informal Care Cost Assessment Questionnaire [48] | 4 | Numerical fields | ✓ | ✗ |
| COVID-19 related | Items related to COVID-19 diagnosis and to assess the impact of COVID-19 on the caregiver's employment, access to support services, willingness to care, provision of practical, emotional and personal care | 15 | Multiple-choice and numerical fields | ✓ | ✗ |
| The use of paid home care | Items asking about the use of paid home care and the demographics of paid care workers | 35 | Yes/no and multiple-choice | ✓ | ✓ |
| Caregiver financial benefits | Items asking about the receipt of cash benefits, financial compensation during care leave, tax benefits, coverage of social or pension contributions, caregiver credits, and health insurance | 6 | Yes/no and numerical fields | ✓ | ✗ |
| Caregiver support services | Items asking about the receipt of caregiver support services | 22 | Yes/no, multiple-choice, and numerical/ text fields | ✓ | ✗ |
| Motivations to provide care | Motivations in Elder Care Scale (MECS) [49] | 13 | Likert scale | ✓ | ✗ |
| Communal motivation to care | Partner-Specific Communal Motivation Scale (CMS) [50] | 10 | Likert scale | ✓ | ✓ (adapted) |
| Willingness and ability to provide care | Willingness to Care Scale [51] | 30 | Yes/no and Likert scale | ✓ | ✗ |
| Willingness to receive care | Items to assess the willingness to receive care. Adapted from Abell [51] | 3 | Likert scale | ✗ | ✓ |
| Well-being | The World Health Organisation-Five Well-Being Index (WHO-5) [52] | 5 | Likert scale | ✓ | ✓ |
| Perceived gains associated with caregiving | The GAINS Scale [53] | 10 | Likert scale | ✓ | ✗ |
| Caregiver burden | Short-Form Zarit Burden Interview (ZBI-12) [54] | 12 | Likert scale | ✓ | ✗ |

(*Continued*)

**Table 1.** (Continued)

| Module/Topic | Measure(s) | Max number of items | Answer type | CG[a] Baseline Survey | CR[b] Baseline Survey |
|---|---|---|---|---|---|
| Health-related quality of life | EuroQol-5D (EQ-5D-5L) [55] | 6 | Likert scale | ✓ | ✓ |
| Depression | Centre for Epidemiological Studies Depression Scale (CESD-10) [56,57] | 10 | Likert scale | ✓ | ✓ |
| Positive dyadic interactions and negative dyadic strain | Dyadic Relationship Scale (DRS) [58] | 11 | Likert scale | ✓ | ✓ (10 items) |
| Relationship satisfaction | Relationship satisfaction (RAS) [59] | 1 | Likert scale | ✓ | ✓ |
| **Module 1: Cultural and psychosocial aspects** | | | | | |
| Familism | Revised Familism Scale (RFS) [60] | 21 | Likert scale | ✓ | ✓ |
| Cognitive and emotional representations of illness. | Brief Illness Perception Questionnaire (B-IPQ) [61] | 9 | Likert scale | ✓ | ✓ |
| Meaning in life | Meaning in Life Questionnaire (MLQ) [62] | 5 | Likert scale | ✓ | ✓ |
| Personal values | Portrait Values Questionnaire (PVQ-21) [63] | 9 | Likert scale | ✓ | ✓ |
| Perceived choice in assuming the caregiving role | Single item: Do you feel you had a choice in taking on this responsibility of caring for your loved one? | 1 | Yes/no | ✓ | ✗ |
| **Module 2: Personality and geographical barriers** | | | | | |
| Geographic-related questions | Items concerning care setting and access, and perceived geographical barriers and facilitators to informal care provision | 30 | Yes/no, multiple-choice, and numerical/ text fields | ✓ | ✓ (8 items) |
| Personality | Big-Five Inventory Extra Short Form (BFI-2-XS) [64] | 15 | Likert scale | ✓ | ✓ |
| Attachment patterns | The Relationship Structures Questionnaire of the Experiences in Close Relationships—Revised (ECR-RS) [65] | 9 | Likert scale | ✓ | ✓ |
| Empathy | Toronto Empathy Questionnaire (TEQ) [66] | 16 | Likert scale | ✓ | ✓ |
| Individuals' personal mastery | The Pearlin Mastery Scale [67] | 7 | Likert scale | ✓ | ✓ |
| **Module 3: Interpersonal processes** | | | | | |
| Collaboration between caregiver and care recipient | Perception of Collaboration Questionnaire (PCQ) [68] | 9 | Likert scale | ✓ | ✓ |
| Perceived communication and dyadic coping within a close relationship when one or both dyad members are stressed | Dyadic Coping Inventory (DCI)-communication subscale [69] | 8 | Likert scale | ✓ | ✓ |
| Mutuality across four dimensions: love and affection, shared pleasurable activities, shared values, and reciprocity | Mutuality Scale (MS) [70] | 15 | Likert scale | ✓ | ✓ |
| Perceived partner responsiveness | The perceived partner responsiveness scale (PPRS) [71] | 12 | Likert scale | ✓ | ✓ |
| Perceived (un)supportive behaviours | Social Support List (SSL) [72] | 13 | Likert scale | ✓ | ✓ |
| **Module 4: Employment, costs, and migrant care work** | | | | | |
| The impact of informal care on employment | Caregiver Indirect and Informal Care Cost Assessment Questionnaire [48] | 7 | Yes/no, numerical fields, and Likert scale | ✓ | ✗ |
| Types of home care services provided by paid care workers | Items asking which tasks and how many hours of care tasks (total and per type of care task) are provided by paid home care workers | 18 | Yes/no, and numerical fields | ✓ | ✗ |
| The rationale for the hiring of paid care workers | Items assess the rationale for hiring paid home care workers and the decision to hire or not hire migrant care workers | 22 | Yes/no | ✓ | ✓ |

(*Continued*)

**Table 1.** (Continued)

| Module/Topic | Measure(s) | Max number of items | Answer type | CG[a] Baseline Survey | CR[b] Baseline Survey |
|---|---|---|---|---|---|
| Out-of-pocket expenses incurred due to caregiving provision | Items to measure out-of-pocket costs (both in terms of the overall total and per type of cost) incurred by caregivers | 25 | Yes/no, and numerical fields | ✓ | ✗ |
| Care benefits received by the care recipient | Items to measure care benefits received (both in terms of the overall total and per type of care benefit) | 4 | Yes/no, and numerical fields | ✗ | ✓ |
| The use of, and the out-of-pocket expenses for, care services, as well as assistive devices and aids used by the care recipient | Items concerning the types of services the care recipient receives in relation to their care and the out-of-pocket expenses spent in relation to their care | 22 | Yes/no, and numerical fields | ✗ | ✓ |

[a] Caregiver.

[b] CR = Care recipient.

calculated by adding the points for each item [53]. Caregiver burden was measured using the Short-Form Zarit Burden Interview (ZBI-12). This instrument includes 12 items, each rated on a 6-point Likert scale from 0 ("never") to 5 ("nearly always"). The total score was obtained by summing the ratings of each item, resulting in a possible range from 0 to 60, with higher scores indicating higher levels of burden [54].

The well-being of caregivers and care recipients was evaluated using the World Health Organisation-Five Well-Being Index (WHO-5). The instrument consists of five positively worded statements related to well-being, each scored on a scale of 0 ("at no time") to 5 ("all of the time"). The sum of the scores for the five items (raw score) is multiplied by four, resulting in a percentage score ranging from 0 (worst imaginable well-being) to 100 (highest imaginable well-being) [52].

**Translation and piloting of the surveys.** All study surveys were first developed in English and then translated into the local languages of the participating countries (Polish, Italian, Dutch, Swedish, Greek, Hebrew and German) using the forward-backward translation approach [74]. Validated translations of questionnaires and scales were used wherever available, and those not available in the local languages of participating countries were translated after permission from the authors. The forward translation was conducted by professional bilingual translators who were native speakers of the target languages and fluent in English. The forward translation was translated back into English by independent professional bilingual translators who did not participate in the forward translation process. The forward and backward translations were then reviewed by members of the research team who are native in the target languages, and all identified inconsistencies were resolved in consultation with the translators involved. Before agreement on the final versions, web-based versions of the translated surveys were created and then tested and piloted in two phases.

The surveys were tested in-house by the research team and ENTWINE consortium members whose mother tongue is the target language. The purpose of the testing phase was twofold. First, all testers (n = 17) were asked to check the technical functionality of the surveys. This included accessibility, readability, formatting, questions' routing, response validation features, and data collection procedures (e.g., invitations and reminders). Second, testers were asked to provide feedback on the translation, time to complete the survey, questions' comprehension and flow, and language consistency. This phase did not uncover any major technical problems, and the surveys functioned as intended on all devices and browsers tested. Testers' comments on the content of the surveys were minor, and changes and refinements were made

accordingly. Second, the surveys were pilot-tested using a convenience sample of 25 caregivers and 21 care recipients. Pilot participants were given the opportunity to provide typed feedback on the clarity and understandability of the questions and on any particular problems encountered in completing the surveys. Pilot data were analysed to check for any abnormal patterns of response (e.g., straight-lining) and non-response and to assess the surveys' length and the quality of free-text responses. Based on pilot feedback and data, the research team further refined the surveys for clarity, and time and ease of completion. These refinements included rewording some items and instructions, shortening the survey by eliminating some questions and using shorter questionnaires, and reducing the number of questions per page to avoid excessive scrolling.

### Ethics and governance

Ethical approvals for the study and its consent procedure were obtained from multiple institutions across the participating countries: Institutional Review Board, Bangor University, The UK; NHS Research Ethics and Governance Committee, The UK (reference number: 20/WA/0006); Central Ethics Review Board non-WMO studies, University Medical Center Groningen, The Netherlands (reference number: 201900810); Bar-Ilan University, Faculty of Social Sciences, Department of Psychology, Ethics Committee, Israel (reference number: 36–20); Commissione Etica per la Ricerca in Psicologia (CERPS), Università Cattolica del Sacro Cuore di Milano, Italy (reference number: 31–20); Swedish Ethical Review Authority, Uppsala University, Sweden (reference number: 2020–04569); and Medical Ethics Committee, University of Oldenburg, Germany (reference number: 2020–155). In Poland, the ethical board of the Institute of Psychology at the University of Wrocław recognised the UK NHS Research Ethics Approval as sufficient for conducting the study. Similarly, in Greece, the Department of Psychology at the University of Crete acknowledged the ethical approval granted by the University Medical Center Groningen on the basis of approvals from other European nations. In Ireland, Care Alliance Ireland, a registered charity, deemed the ethical approval from Bangor University sufficient for participant recruitment and study conduct. All these ethical approvals were obtained before initiating recruitment, enrolment, and data collection. All participants were required to give informed written consent via the survey platform Questback before they could gain access to the surveys. Individuals who did not provide consent were denied participation and redirected to a page thanking them for their interest.

## Findings to date

### Recruitment and inclusion

Baseline recruitment was staggered over the 12-month baseline data collection period from 14 August 2020 to 31 August 2021. The start dates for recruitment varied by country as follows: the UK, Ireland, and Poland on 14 August 2020; Italy on 25 August 2020; the Netherlands on 14 October 2020; Sweden on 23 October 2020; Greece on 31 October 2020; and Germany and Israel on 16 February 2021. Baseline recruitment was concluded on 31 May 2021 in all countries except Germany and Israel. In these two countries, recruitment was extended until 31 August 2021, thereby ensuring a recruitment period of at least six months in all participating countries.

Fig 1 illustrates the dates for key recruitment activities and the number of participants enrolled in the cohort in each month of baseline recruitment. Overall, recruitment proceeded slowly until 1 December 2020, with a mean of 124 recruited participants per month. Then the Facebook paid campaign was launched for five months (until 1 May 2021), during which the monthly enrolment rate averaged 318 recruited participants per month. The monthly

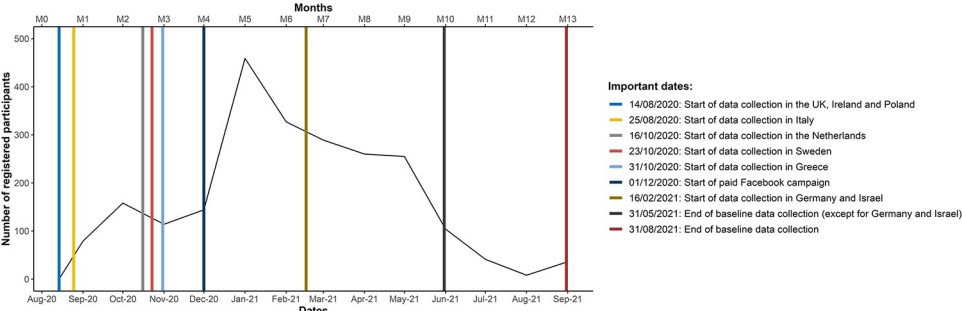

**Fig 1. Number of participants enrolled in the cohort in each month of the baseline recruitment period.**

enrolment rate for the rest of the recruitment period (1 May 2021 through 31 August 2021) was 48 participants per month. The intended duration for baseline recruitment was six months, but it was decided to extend it for six months to recruit a minimum of 2000 participants.

By the end of baseline recruitment, the link to the eligibility survey had 149,129 hits. The number of visitors who completed at least the first question of the eligibility survey was 24,945 (16.7%), of which 18,685 (75.0%) were excluded due to not completing the survey further, consent withdrawal, or missing the deadline of data collection in their respective country of residence. Out of the 6,260 respondents who completed the eligibility survey, 2,893 (46.2%) did not meet the study's eligibility criteria. This left 3,367 participants available for the study, comprising 2,731 caregivers (81.1%) and 636 care recipients (18.9%). At baseline, 859 caregivers (31.5%) and 234 care recipients (36.8%) either did not access the baseline survey on receipt of the link sent or could not be contacted (i.e., provided incorrect email address). Thus, a total of 1,872 caregivers and 402 care recipients were enrolled in the cohort. The flow chart of recruitment is shown in Fig 2.

The number of unique visitors to the eligibility survey could not be determined, as IP addresses were not recorded and cookies were not used, as stated above. Therefore, response and completion rates could not be computed for the eligibility survey. For the Caregiver and Care Recipient baseline surveys, the response rate, defined here as the number of fully completed surveys by the number of invitation emails sent, was 42% and 43%, respectively [75]. The completion rate (i.e., the number of fully completed surveys by the number of participants who accessed the survey) for the Caregiver and Care Recipient baseline surveys was 61% and 69%, respectively [75].

The majority of the ENTWINE iCohort were recruited through social media (n = 1,664, 73.2%), especially Facebook. Each of the other recruitment methods yielded less than 8% of the total cohort sample. Fig 3 displays the number and percentage of participants enrolled by each recruitment method.

### Baseline cohort characteristics

**Characteristics of caregiver participants.** The baseline characteristics of caregiver participants and their care recipients overall and by country are presented in Table 2. The majority of caregiver participants in all countries were females (87.2%). The median age of the caregiver sample was 55 years, with the youngest age in Poland (50y), Greece, Israel, and Italy (53y) and the oldest in Ireland (56y), the Netherlands (57y), the UK (58y), and Sweden (61y). Nearly two-thirds of caregiver participants (66.7%) indicated being married or in a partnership, with some variation across countries (54.4% in Greece to 73.7% in Sweden). Across all countries,

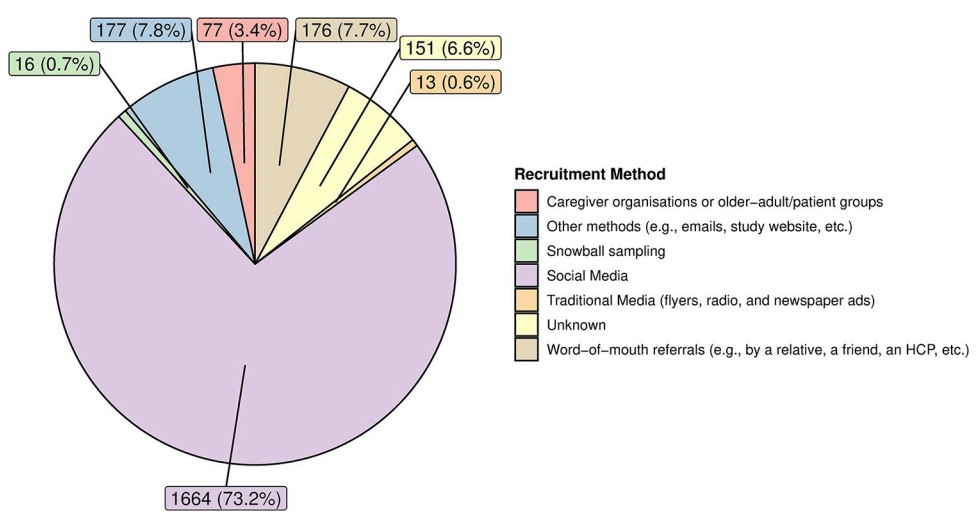

**Fig 2. Flow chart of baseline recruitment for the ENTWINE iCohort Study.**

**Fig 3. Number and percentage of participants enrolled by each recruitment method.**

**Table 2. Baseline characteristics of caregiver participants and their care recipients overall and by country.**

| Characteristic | N | Germany, N = 38 | Greece, N = 160 | Ireland, N = 91 | Israel, N = 215 | Italy, N = 391 | Netherlands, N = 421 | Poland, N = 171 | Sweden, N = 156 | UK, N = 229 | Overall, N = 1,872 |
|---|---|---|---|---|---|---|---|---|---|---|---|
| **Age, Median (SD)** | 1872 | 59.0 | 53.0 | 56.0 | 53.0 | 53.0 | 57.0 | 50.0 | 61.0 | 58.0 | 55.0 |
| | | (13.5) | (11.5) | (11.2) | (15.6) | (11.7) | (11.1) | (12.9) | (13.2) | (12.4) | (12.8) |
| **Gender, n (%)** | 1872 | | | | | | | | | | |
| _Female_ | | 28 | 138 | 81 | 169 | 336 | 385 | 154 | 137 | 205 | 1,633 |
| | | (73.7%) | (86.2%) | (89.0%) | (78.6%) | (85.9%) | (91.4%) | (90.1%) | (87.8%) | (89.5%) | (87.2%) |
| _Male_ | | 9 | 21 | 7 | 45 | 50 | 33 | 17 | 19 | 23 | 224 |
| | | (23.7%) | (13.1%) | (7.7%) | (20.9%) | (12.8%) | (7.8%) | (9.9%) | (12.2%) | (10.0%) | (12.0%) |
| _Non-binary/third gender_ | | 0 | 1 | 0 | 0 | 1 | 1 | 0 | 0 | 1 | 4 |
| | | (0.0%) | (0.6%) | (0.0%) | (0.0%) | (0.3%) | (0.2%) | (0.0%) | (0.0%) | (0.4%) | (0.2%) |
| _Prefer to self-describe_ | | 0 | 0 | 2 | 1 | 1 | 2 | 0 | 0 | 0 | 6 |
| | | (0.0%) | (0.0%) | (2.2%) | (0.5%) | (0.3%) | (0.5%) | (0.0%) | (0.0%) | (0.0%) | (0.3%) |
| _Prefer not to say_ | | 1 | 0 | 1 | 0 | 3 | 0 | 0 | 0 | 0 | 5 |
| | | (2.6%) | (0.0%) | (1.1%) | (0.0%) | (0.8%) | (0.0%) | (0.0%) | (0.0%) | (0.0%) | (0.3%) |
| **Marital status, n (%)** | 1872 | | | | | | | | | | |
| _Single_ | | 6 | 36 | 14 | 47 | 104 | 38 | 39 | 12 | 22 | 318 |
| | | (15.8%) | (22.5%) | (15.4%) | (21.9%) | (26.6%) | (9.0%) | (22.8%) | (7.7%) | (9.6%) | (17.0%) |
| _Married or in a partnership_ | | 26 | 87 | 65 | 141 | 243 | 301 | 102 | 115 | 168 | 1,248 |
| | | (68.4%) | (54.4%) | (71.4%) | (65.6%) | (62.1%) | (71.5%) | (59.6%) | (73.7%) | (73.4%) | (66.7%) |
| _Divorced_ | | 6 | 23 | 7 | 21 | 25 | 37 | 18 | 18 | 26 | 181 |
| | | (15.8%) | (14.4%) | (7.7%) | (9.8%) | (6.4%) | (8.8%) | (10.5%) | (11.5%) | (11.4%) | (9.7%) |
| _Widowed_ | | 0 | 10 | 1 | 4 | 7 | 11 | 8 | 6 | 9 | 56 |
| | | (0.0%) | (6.2%) | (1.1%) | (1.9%) | (1.8%) | (2.6%) | (4.7%) | (3.8%) | (3.9%) | (3.0%) |
| _Other_ | | 0 | 4 | 4 | 2 | 12 | 34 | 4 | 5 | 4 | 69 |
| | | (0.0%) | (2.5%) | (4.4%) | (0.9%) | (3.1%) | (8.1%) | (2.3%) | (3.2%) | (1.7%) | (3.7%) |
| **Employed = Yes, n (%)** | 1837 | 20 | 90 | 30 | 145 | 199 | 224 | 99 | 68 | 91 | 966 |
| | | (54.1%) | (57.3%) | (34.1%) | (68.1%) | (51.7%) | (54.5%) | (60.4%) | (43.6%) | (40.3%) | (52.6%) |
| _Missing_ | | 1 | 3 | 3 | 2 | 6 | 10 | 7 | 0 | 3 | 35 |
| **Education, n (%)** | 1872 | | | | | | | | | | |
| _Primary_ | | 0 | 5 | 4 | 1 | 2 | 5 | 0 | 8 | 1 | 26 |
| | | (0.0%) | (3.1%) | (4.4%) | (0.5%) | (0.5%) | (1.2%) | (0.0%) | (5.1%) | (0.4%) | (1.4%) |
| _Secondary_ | | 2 | 37 | 13 | 36 | 29 | 68 | 17 | 24 | 38 | 264 |
| | | (5.3%) | (23.1%) | (14.3%) | (16.7%) | (7.4%) | (16.2%) | (9.9%) | (15.4%) | (16.6%) | (14.1%) |
| _Post-secondary vocational education_ | | 19 | 41 | 20 | 31 | 185 | 298 | 34 | 34 | 64 | 726 |
| | | (50.0%) | (25.6%) | (22.0%) | (14.4%) | (47.3%) | (70.8%) | (19.9%) | (21.8%) | (27.9%) | (38.8%) |
| _Post-secondary academic education_ | | 14 | 75 | 53 | 145 | 173 | 42 | 105 | 86 | 124 | 817 |
| | | (36.8%) | (46.9%) | (58.2%) | (67.4%) | (44.2%) | (10.0%) | (61.4%) | (55.1%) | (54.1%) | (43.6%) |
| _Not listed or other_ | | 3 | 2 | 1 | 2 | 2 | 8 | 15 | 4 | 2 | 39 |
| | | (7.9%) | (1.2%) | (1.1%) | (0.9%) | (0.5%) | (1.9%) | (8.8%) | (2.6%) | (0.9%) | (2.1%) |
| **Primary caregiver = Yes, n (%)** | 1581 | 29 | 103 | 69 | 119 | 257 | 307 | 107 | 130 | 179 | 1,300 |
| | | (78.4%) | (74.1%) | (93.2%) | (64.0%) | (79.1%) | (88.5%) | (82.3%) | (90.9%) | (89.5%) | (82.2%) |
| _Missing_ | | 1 | 21 | 17 | 29 | 66 | 74 | 41 | 13 | 29 | 291 |
| **Total weekly hours of care, Mean (SD)** | 1604 | 59.8 | 47.2 | 71.7 | 28.9 | 64.6 | 38.3 | 61.6 | 40.6 | 61.4 | 50.5 |
| | | (41.3) | (35.1) | (40.8) | (31.5) | (41.2) | (35.2) | (39.8) | (32.8) | (39.9) | (39.7) |
| _Missing_ | | 1 | 19 | 16 | 26 | 62 | 68 | 39 | 12 | 25 | 268 |
| **WHO-5 score[a], Mean (SD)** | 1368 | 48.9 | 43.5 | 39.9 | 60.7 | 39.3 | 53.1 | 31.8 | 43.4 | 43.5 | 45.8 |
| | | (22.4) | (24.0) | (23.0) | (22.8) | (22.7) | (24.0) | (21.5) | (24.2) | (23.4) | (24.5) |

_(Continued)_

**Table 2.** (Continued)

| Characteristic | N | Germany, N = 38 | Greece, N = 160 | Ireland, N = 91 | Israel, N = 215 | Italy, N = 391 | Netherlands, N = 421 | Poland, N = 171 | Sweden, N = 156 | UK, N = 229 | Overall, N = 1,872 |
|---|---|---|---|---|---|---|---|---|---|---|---|
| *Score ≤ 50[b], n (%)* | | 16 | 74 | 44 | 53 | 189 | 127 | 84 | 75 | 113 | 775 |
| | | (44.4%) | (60.2%) | (67.7%) | (31.7%) | (67.3%) | (44.9%) | (79.2%) | (60.5%) | (61.7%) | (56.7%) |
| *Missing* | | 2 | 37 | 26 | 48 | 110 | 138 | 65 | 32 | 46 | 504 |
| **Zarit Burden Interview score, Mean (SD)** | 1340 | 20.0 | 22.3 | 22.0 | 16.5 | 21.4 | 17.9 | 22.7 | 24.0 | 22.5 | 20.6 |
| | | (7.9) | (9.6) | (9.7) | (8.6) | (8.5) | (9.0) | (8.6) | (9.7) | (9.4) | (9.3) |
| *Missing* | | 2 | 37 | 28 | 50 | 119 | 141 | 69 | 35 | 51 | 532 |
| **GAINS score, Mean (SD)** | 1344 | 16.1 | 10.4 | 13.2 | 10.5 | 10.5 | 14.4 | 11.9 | 15.6 | 16.2 | 12.9 |
| | | (6.1) | (6.7) | (5.8) | (6.4) | (6.3) | (6.4) | (6.0) | (6.1) | (6.5) | (6.7) |
| *Missing* | | 2 | 37 | 27 | 51 | 118 | 141 | 67 | 33 | 52 | 528 |
| **Willingness to Care score, Mean (SD)** | 1309 | 4.0 | 4.3 | 4.5 | 4.2 | 4.4 | 4.4 | 4.0 | 3.8 | 4.5 | 4.3 |
| | | (0.8) | (0.7) | (0.6) | (0.7) | (0.6) | (0.7) | (0.7) | (0.6) | (0.5) | (0.7) |
| *Missing* | | 5 | 40 | 36 | 49 | 129 | 146 | 76 | 34 | 48 | 563 |
| **Ability to Care score, Mean (SD)** | 1304 | 26.1 | 28.0 | 28.6 | 26.5 | 28.2 | 27.4 | 28.2 | 27.2 | 28.2 | 27.7 |
| | | (4.8) | (3.1) | (2.4) | (4.3) | (2.6) | (3.9) | (3.0) | (3.9) | (3.4) | (3.6) |
| *Missing* | | 5 | 41 | 37 | 48 | 129 | 145 | 75 | 35 | 53 | 568 |
| **Caregiver's relationship to care recipient, n (%)** | 1724 | | | | | | | | | | |
| *Spouse/Partner* | | 14 | 16 | 28 | 36 | 76 | 165 | 30 | 89 | 90 | 544 |
| | | (37.8%) | (11.0%) | (35.0%) | (17.8%) | (21.3%) | (43.2%) | (19.6%) | (58.6%) | (41.7%) | (31.6%) |
| *Mother or father* | | 18 | 98 | 25 | 104 | 189 | 114 | 80 | 29 | 73 | 730 |
| | | (48.6%) | (67.1%) | (31.2%) | (51.5%) | (53.1%) | (29.8%) | (52.3%) | (19.1%) | (33.8%) | (42.3%) |
| *Mother-in-law or father-in-law* | | 1 | 3 | 3 | 10 | 7 | 11 | 4 | 1 | 5 | 45 |
| | | (2.7%) | (2.1%) | (3.8%) | (5.0%) | (2.0%) | (2.9%) | (2.6%) | (0.7%) | (2.3%) | (2.6%) |
| *Daughter or son* | | 1 | 8 | 15 | 9 | 42 | 43 | 10 | 26 | 26 | 180 |
| | | (2.7%) | (5.5%) | (18.8%) | (4.5%) | (11.8%) | (11.3%) | (6.5%) | (17.1%) | (12.0%) | (10.4%) |
| *Grandmother/Grandfather* | | 2 | 2 | 0 | 19 | 10 | 3 | 8 | 0 | 1 | 45 |
| | | (5.4%) | (1.4%) | (0.0%) | (9.4%) | (2.8%) | (0.8%) | (5.2%) | (0.0%) | (0.5%) | (2.6%) |
| *Sibling* | | 0 | 7 | 3 | 11 | 11 | 6 | 6 | 4 | 7 | 55 |
| | | (0.0%) | (4.8%) | (3.8%) | (5.4%) | (3.1%) | (1.6%) | (3.9%) | (2.6%) | (3.2%) | (3.2%) |
| *Another family member* | | 1 | 1 | 0 | 5 | 6 | 12 | 4 | 0 | 5 | 34 |
| | | (2.7%) | (0.7%) | (0.0%) | (2.5%) | (1.7%) | (3.1%) | (2.6%) | (0.0%) | (2.3%) | (2.0%) |
| *Friend* | | 0 | 4 | 3 | 1 | 4 | 9 | 4 | 0 | 4 | 29 |
| | | (0.0%) | (2.7%) | (3.8%) | (0.5%) | (1.1%) | (2.4%) | (2.6%) | (0.0%) | (1.9%) | (1.7%) |
| *Acquaintance / Neighbour / Other non-relative* | | 0 | 7 | 1 | 2 | 6 | 9 | 4 | 1 | 0 | 30 |
| | | (0.0%) | (4.8%) | (1.2%) | (1.0%) | (1.7%) | (2.4%) | (2.6%) | (0.7%) | (0.0%) | (1.7%) |
| *Other* | | 0 | 0 | 2 | 5 | 5 | 10 | 3 | 2 | 5 | 32 |
| | | (0.0%) | (0.0%) | (2.5%) | (2.5%) | (1.4%) | (2.6%) | (2.0%) | (1.3%) | (2.3%) | (1.9%) |
| *Missing* | | 1 | 14 | 11 | 13 | 35 | 39 | 18 | 4 | 13 | 148 |
| **Care recipient's age, Median (SD)** | 1738 | 77.0 | 80.0 | 69.0 | 78.0 | 75.0 | 69.0 | 77.0 | 70.0 | 72.0 | 73.0 |
| | | (11.4) | (18.2) | (22.4) | (19.4) | (20.8) | (19.6) | (17.7) | (19.5) | (20.2) | (19.9) |
| *Missing* | | 1 | 14 | 11 | 10 | 31 | 33 | 18 | 4 | 12 | 134 |
| **Care recipient's gender, n (%)** | 1738 | | | | | | | | | | |
| *Female* | | 16 | 105 | 31 | 120 | 205 | 158 | 95 | 51 | 90 | 871 |
| | | (43.2%) | (71.9%) | (38.8%) | (58.5%) | (56.9%) | (40.7%) | (62.1%) | (33.6%) | (41.5%) | (50.1%) |
| *Male* | | 21 | 40 | 49 | 84 | 154 | 227 | 56 | 100 | 126 | 857 |
| | | (56.8%) | (27.4%) | (61.3%) | (41.0%) | (42.8%) | (58.5%) | (36.6%) | (65.8%) | (58.1%) | (49.3%) |

*(Continued)*

**Table 2.** (Continued)

| Characteristic | N | Germany, N = 38 | Greece, N = 160 | Ireland, N = 91 | Israel, N = 215 | Italy, N = 391 | Netherlands, N = 421 | Poland, N = 171 | Sweden, N = 156 | UK, N = 229 | Overall, N = 1,872 |
|---|---|---|---|---|---|---|---|---|---|---|---|
| *Prefer to self-describe* | | 0 | 0 | 0 | 1 | 0 | 3 | 1 | 0 | 0 | 5 |
| | | (0.0%) | (0.0%) | (0.0%) | (0.5%) | (0.0%) | (0.8%) | (0.7%) | (0.0%) | (0.0%) | (0.3%) |
| *Prefer not to say* | | 0 | 1 | 0 | 0 | 1 | 0 | 1 | 1 | 1 | 5 |
| | | (0.0%) | (0.7%) | (0.0%) | (0.0%) | (0.3%) | (0.0%) | (0.7%) | (0.7%) | (0.5%) | (0.3%) |
| *Missing* | | 1 | 14 | 11 | 10 | 31 | 33 | 18 | 4 | 12 | 134 |
| **Katz ADL Index score of the care recipient[c], Median (SD)** | 1713 | 3.0 | 3.0 | 3.0 | 4.0 | 1.0 | 4.0 | 2.0 | 4.0 | 2.0 | 3.0 |
| | | (2.0) | (2.2) | (2.2) | (2.4) | (2.2) | (2.2) | (2.3) | (2.3) | (2.1) | (2.3) |
| *Missing* | | 1 | 14 | 11 | 13 | 40 | 43 | 19 | 4 | 14 | 159 |
| **Dependency level of the care recipient[d], n (%)** | 1713 | | | | | | | | | | |
| *Independent* | | 4 | 27 | 13 | 68 | 55 | 111 | 24 | 55 | 39 | 396 |
| | | (10.8%) | (18.5%) | (16.2%) | (33.7%) | (15.7%) | (29.4%) | (15.8%) | (36.2%) | (18.1%) | (23.1%) |
| *Partially Dependent* | | 18 | 56 | 34 | 64 | 82 | 137 | 47 | 50 | 64 | 552 |
| | | (48.6%) | (38.4%) | (42.5%) | (31.7%) | (23.4%) | (36.2%) | (30.9%) | (32.9%) | (29.8%) | (32.2%) |
| *Dependent* | | 15 | 63 | 33 | 70 | 214 | 130 | 81 | 47 | 112 | 765 |
| | | (40.5%) | (43.2%) | (41.2%) | (34.7%) | (61.0%) | (34.4%) | (53.3%) | (30.9%) | (52.1%) | (44.7%) |
| *Missing* | | 1 | 14 | 11 | 13 | 40 | 43 | 19 | 4 | 14 | 159 |

WHO-5: The World Health Organisation Five Well-Being Index; ADL: Activities of Daily Living (eating, bathing, dressing, toileting, transferring, and continence).

[a]The range of score is from 0 (worst imaginable well-being) to 100 (highest imaginable well-being).

[b]WHO-5 score $\leq$ 50 indicates suboptimal well-being [76].

[c]Katz ADL index score of the care recipient as reported by the caregiver. The score ranges from 0 (dependence in all ADL) to 6 (total independence).

[d]Dependency level according to the Katz ADL index score of the care recipient as reported by the caregiver: Independent (score 6), Partially dependent (score 3–5), and Dependent (score 0–2) [73].

more than half of the participants (52.6%) were employed, with the percentages somewhat lower in Ireland (34.1%), the UK (40.3%), and Sweden (43.6%). More than 80% of participants held at least post-secondary education (i.e., tertiary vocational or academic education). The majority of caregiver participants reported being the primary caregiver for the care recipient they cared for (82.2%) and providing, on average, 50.5 hours of care per week.

The mean WHO-5 score, reflecting caregivers' well-being, was 45.8, with 56.7% of caregivers reporting suboptimal well-being (using a cut-off score of 50 [76]). Caregivers, on average, scored 20.6 on ZBI-12, indicating that they experience moderate to high burden, closer to the moderate end. The mean score of the GAINS scale was 12.9 out of a possible 30, with higher scores indicative of greater perceived gains from caregiving. Regarding the willingness and ability to care, the mean scores were 4.3 (max score = 5) and 27.7 (max score = 30), respectively (higher scores indicative of greater willingness and ability to care).

Caregiver participants were providing care for a parent (42.3%), partner (31.6%), child (10.4%), or sibling (3.2%), with the remaining (12.5%) providing care for other family members and non-relatives. About half of participants' care recipients were females (50.1%), and their median age was 73 years, with the youngest in Ireland and the Netherlands (69y), Sweden (70y) and the UK (72y), and the oldest in Italy (75y), Germany and Poland (77y), Israel (78y) and Greece (80y). According to the Katz index (as reported by the caregiver participant), 76.9% of participants' care recipients were found to be at least partially dependent for ADL (Mdn = 3). With regards to the health conditions of participants' care recipients, the most reported condition by caregivers was other (32.5%), which represented conditions that did not

fit into any other category (e.g., genetic disorders, physical disabilities, and mental health conditions). The next most reported condition was cognitive or memory impairment (29.9%), followed by hypertension (26.3%), heart disease (16.9%), diabetes (16.8%), stroke or cerebral vascular disease (16.0%), cancer (15.7%), high blood cholesterol (13.7%), cataracts (10.4%), and osteoarthritis or other rheumatism (10.3%). Each of the remaining conditions was reported in less than 10% of participants' care recipients. Also, 55.8% of caregivers reported providing care for a person with multiple chronic health conditions. Only 3.6% of caregivers reported that their care recipients were not diagnosed with any chronic disease, relating instead to care provided due to old age frailty or following an injury or acute illness. S2 Table presents the chronic conditions of the care recipients as reported by the caregivers.

**Characteristics of care recipient participants.** Table 3 presents the baseline characteristics of care recipient participants overall and by country. Care recipients had a median age of

**Table 3. Baseline characteristics of care recipient participants overall and by country.**

| Characteristic | N | Germany, N = 7 | Greece, N = 37 | Ireland, N = 26 | Israel, N = 26 | Italy, N = 67 | Netherlands, N = 101 | Poland, N = 46 | Sweden, N = 26 | UK, N = 66 | Overall, N = 402 |
|---|---|---|---|---|---|---|---|---|---|---|---|
| **Age, Median (SD)** | 402 | 56.0 | 46.0 | 59.5 | 49.0 | 47.0 | 58.0 | 52.5 | 66.0 | 59.5 | 55.0 |
| | | (14.5) | (12.7) | (14.2) | (17.8) | (15.8) | (12.6) | (14.9) | (15.5) | (12.9) | (14.9) |
| **Gender, n (%)** | 402 | | | | | | | | | | |
| _Female_ | | 6 | 27 | 19 | 21 | 46 | 82 | 33 | 19 | 57 | 310 |
| | | (85.7%) | (73.0%) | (73.1%) | (80.8%) | (68.7%) | (81.2%) | (71.7%) | (73.1%) | (86.4%) | (77.1%) |
| _Male_ | | 1 | 9 | 7 | 3 | 20 | 19 | 13 | 6 | 9 | 87 |
| | | (14.3%) | (24.3%) | (26.9%) | (11.5%) | (29.9%) | (18.8%) | (28.3%) | (23.1%) | (13.6%) | (21.6%) |
| _Non-binary/third gender_ | | 0 | 0 | 0 | 1 | 0 | 0 | 0 | 0 | 0 | 1 |
| | | (0.0%) | (0.0%) | (0.0%) | (3.8%) | (0.0%) | (0.0%) | (0.0%) | (0.0%) | (0.0%) | (0.2%) |
| _Prefer to self-describe_ | | 0 | 0 | 0 | 1 | 0 | 0 | 0 | 0 | 0 | 1 |
| | | (0.0%) | (0.0%) | (0.0%) | (3.8%) | (0.0%) | (0.0%) | (0.0%) | (0.0%) | (0.0%) | (0.2%) |
| _Prefer not to say_ | | 0 | 1 | 0 | 0 | 1 | 0 | 0 | 1 | 0 | 3 |
| | | (0.0%) | (2.7%) | (0.0%) | (0.0%) | (1.5%) | (0.0%) | (0.0%) | (3.8%) | (0.0%) | (0.7%) |
| **Education, n (%)** | 402 | | | | | | | | | | |
| _Primary_ | | 1 | 0 | 1 | 0 | 2 | 4 | 1 | 2 | 2 | 13 |
| | | (14.3%) | (0.0%) | (3.8%) | (0.0%) | (3.0%) | (4.0%) | (2.2%) | (7.7%) | (3.0%) | (3.2%) |
| _Secondary_ | | 0 | 13 | 3 | 5 | 4 | 18 | 9 | 5 | 10 | 67 |
| | | (0.0%) | (35.1%) | (11.5%) | (19.2%) | (6.0%) | (17.8%) | (19.6%) | (19.2%) | (15.2%) | (16.7%) |
| _Post-secondary vocational education_ | | 4 | 10 | 8 | 6 | 35 | 68 | 14 | 4 | 22 | 171 |
| | | (57.1%) | (27.0%) | (30.8%) | (23.1%) | (52.2%) | (67.3%) | (30.4%) | (15.4%) | (33.3%) | (42.5%) |
| _Post-secondary academic education_ | | 1 | 14 | 14 | 14 | 26 | 9 | 19 | 14 | 28 | 139 |
| | | (14.3%) | (37.8%) | (53.8%) | (53.8%) | (38.8%) | (8.9%) | (41.3%) | (53.8%) | (42.4%) | (34.6%) |
| _Not listed or other_ | | 1 | 0 | 0 | 1 | 0 | 2 | 3 | 1 | 4 | 12 |
| | | (14.3%) | (0.0%) | (0.0%) | (3.8%) | (0.0%) | (2.0%) | (6.5%) | (3.8%) | (6.1%) | (3.0%) |
| **Employed = Yes, n (%)** | 400 | 0 | 16 | 6 | 9 | 28 | 16 | 14 | 5 | 10 | 104 |
| | | (0.0%) | (43.2%) | (23.1%) | (34.6%) | (41.8%) | (16.0%) | (31.1%) | (19.2%) | (15.2%) | (26.0%) |
| _Missing_ | | 0 | 0 | 0 | 0 | 0 | 1 | 1 | 0 | 0 | 2 |
| **Marital status, n (%)** | 402 | | | | | | | | | | |
| _Single_ | | 1 | 12 | 5 | 9 | 27 | 10 | 9 | 2 | 10 | 85 |
| | | (14.3%) | (32.4%) | (19.2%) | (34.6%) | (40.3%) | (9.9%) | (19.6%) | (7.7%) | (15.2%) | (21.1%) |
| _Married or in a partnership_ | | 4 | 15 | 16 | 13 | 33 | 63 | 29 | 19 | 43 | 235 |
| | | (57.1%) | (40.5%) | (61.5%) | (50.0%) | (49.3%) | (62.4%) | (63.0%) | (73.1%) | (65.2%) | (58.5%) |

_(Continued)_

**Table 3.** (Continued)

| Characteristic | N | Germany, N = 7 | Greece, N = 37 | Ireland, N = 26 | Israel, N = 26 | Italy, N = 67 | Netherlands, N = 101 | Poland, N = 46 | Sweden, N = 26 | UK, N = 66 | Overall, N = 402 |
|---|---|---|---|---|---|---|---|---|---|---|---|
| *Divorced* | | 1 | 7 | 3 | 4 | 4 | 10 | 4 | 3 | 9 | 45 |
| | | (14.3%) | (18.9%) | (11.5%) | (15.4%) | (6.0%) | (9.9%) | (8.7%) | (11.5%) | (13.6%) | (11.2%) |
| *Widowed* | | 1 | 1 | 1 | 0 | 3 | 9 | 4 | 1 | 3 | 23 |
| | | (14.3%) | (2.7%) | (3.8%) | (0.0%) | (4.5%) | (8.9%) | (8.7%) | (3.8%) | (4.5%) | (5.7%) |
| *Other* | | 0 | 2 | 1 | 0 | 0 | 9 | 0 | 1 | 1 | 14 |
| | | (0.0%) | (5.4%) | (3.8%) | (0.0%) | (0.0%) | (8.9%) | (0.0%) | (3.8%) | (1.5%) | (3.5%) |
| **Main caregiver, n (%)** | 382 | | | | | | | | | | |
| *Spouse/Partner* | | 4 | 18 | 14 | 12 | 31 | 67 | 27 | 19 | 41 | 233 |
| | | (57.1%) | (50.0%) | (58.3%) | (48.0%) | (47.7%) | (70.5%) | (61.4%) | (76.0%) | (67.2%) | (61.0%) |
| *Child* | | 2 | 7 | 1 | 1 | 9 | 13 | 3 | 3 | 8 | 47 |
| | | (28.6%) | (19.4%) | (4.2%) | (4.0%) | (13.8%) | (13.7%) | (6.8%) | (12.0%) | (13.1%) | (12.3%) |
| *Parent* | | 0 | 5 | 3 | 6 | 14 | 2 | 7 | 2 | 4 | 43 |
| | | (0.0%) | (13.9%) | (12.5%) | (24.0%) | (21.5%) | (2.1%) | (15.9%) | (8.0%) | (6.6%) | (11.3%) |
| *Sibling* | | 0 | 2 | 0 | 1 | 5 | 3 | 0 | 1 | 2 | 14 |
| | | (0.0%) | (5.6%) | (0.0%) | (4.0%) | (7.7%) | (3.2%) | (0.0%) | (4.0%) | (3.3%) | (3.7%) |
| *Nephew/Niece* | | 0 | 2 | 0 | 0 | 1 | 0 | 2 | 0 | 0 | 5 |
| | | (0.0%) | (5.6%) | (0.0%) | (0.0%) | (1.5%) | (0.0%) | (4.5%) | (0.0%) | (0.0%) | (1.3%) |
| *Friend* | | 0 | 1 | 2 | 1 | 0 | 8 | 1 | 0 | 3 | 16 |
| | | (0.0%) | (2.8%) | (8.3%) | (4.0%) | (0.0%) | (8.4%) | (2.3%) | (0.0%) | (4.9%) | (4.2%) |
| *Neighbour* | | 0 | 0 | 2 | 0 | 0 | 0 | 3 | 0 | 0 | 5 |
| | | (0.0%) | (0.0%) | (8.3%) | (0.0%) | (0.0%) | (0.0%) | (6.8%) | (0.0%) | (0.0%) | (1.3%) |
| *Other* | | 1 | 1 | 2 | 4 | 5 | 2 | 1 | 0 | 3 | 19 |
| | | (14.3%) | (2.8%) | (8.3%) | (16.0%) | (7.7%) | (2.1%) | (2.3%) | (0.0%) | (4.9%) | (5.0%) |
| *Missing* | | 0 | 1 | 2 | 1 | 2 | 6 | 2 | 1 | 5 | 20 |
| **WHO-5 score[a], Mean (SD)** | 343 | 40.0 | 42.8 | 41.2 | 42.4 | 45.2 | 43.5 | 36.6 | 47.5 | 29.9 | 40.7 |
| | | (28.3) | (24.3) | (20.8) | (22.8) | (21.8) | (22.4) | (22.6) | (24.3) | (22.3) | (23.0) |
| *Score $\leq 50$[b], n (%)* | | 5 | 21 | 15 | 13 | 35 | 47 | 28 | 13 | 43 | 220 |
| | | (71.4%) | (63.6%) | (65.2%) | (65.0%) | (58.3%) | (58.8%) | (68.3%) | (56.5%) | (76.8%) | (64.1%) |
| *Missing* | | 0 | 4 | 3 | 6 | 7 | 21 | 5 | 3 | 10 | 59 |
| **Katz ADL index score[c], Median (SD)** | 383 | 3.0 | 4.5 | 5.0 | 5.0 | 5.0 | 4.0 | 4.0 | 4.0 | 2.0 | 4.0 |
| | | (1.1) | (2.2) | (2.1) | (1.6) | (2.1) | (1.9) | (2.1) | (2.1) | (1.9) | (2.0) |
| *Missing* | | 0 | 1 | 2 | 1 | 2 | 5 | 2 | 1 | 5 | 19 |
| **Dependency level[d], n (%)** | 383 | | | | | | | | | | |
| *Independent* | | 1 | 14 | 11 | 10 | 22 | 25 | 15 | 8 | 8 | 114 |
| | | (14.3%) | (38.9%) | (45.8%) | (40.0%) | (33.8%) | (26.0%) | (34.1%) | (32.0%) | (13.1%) | (29.8%) |
| *Partially Dependent* | | 6 | 12 | 8 | 12 | 26 | 46 | 16 | 9 | 20 | 155 |
| | | (85.7%) | (33.3%) | (33.3%) | (48.0%) | (40.0%) | (47.9%) | (36.4%) | (36.0%) | (32.8%) | (40.5%) |
| *Dependent* | | 0 | 10 | 5 | 3 | 17 | 25 | 13 | 8 | 33 | 114 |
| | | (0.0%) | (27.8%) | (20.8%) | (12.0%) | (26.2%) | (26.0%) | (29.5%) | (32.0%) | (54.1%) | (29.8%) |
| *Missing* | | 0 | 1 | 2 | 1 | 2 | 5 | 2 | 1 | 5 | 19 |

WHO-5: The World Health Organisation Five Well-Being Index; ADL: Activities of Daily Living (eating, bathing, dressing, toileting, transferring, and continence).

[a]The range of score is from 0 (worst imaginable well-being) to 100 (highest imaginable well-being).

[b]WHO-5 score $\leq 50$ indicates suboptimal well-being [76].

[c]The score ranges from 0 (dependence in all ADL) to 6 (total independence).

[d]Dependency level according to the Katz ADL index score: Independent (score 6), Partially dependent (score 3–5), and Dependent (score 0–2 [73]).

55 years, with approximately three-quarters being female (77.1%). The majority of care recipient participants had at least a post-secondary education (77.1%) and were not employed (74.0%). More than half of the care recipients were married or in a partnership (58.5%) and were receiving care primarily from their spouses or partners (61.0%). On average, care recipients scored 40.7 on WHO-5, with 64.1% of them reporting suboptimal well-being (WHO-5 score ≤ 50). According to the Katz index, more than two-thirds of surveyed care recipients (70.3%) were at least partially dependent for ADL, with a median Katz index score of 4. Furthermore, more than half of the care recipients (56.8%) had at least two coexisting chronic conditions. The most reported condition was other (48.6%), followed by hypertension (27.6%), osteoarthritis or other rheumatism (21.7%), cancer (18.9%), high blood cholesterol (18.6%), diabetes (17.6%), chronic lung disease (16.5%), and heart diseases (10.6%), with the remaining conditions contributing less than 10% each. The complete list of reported chronic health conditions and their prevalence among the care recipient sample is shown in S3 Table.

## Published studies and ongoing projects

One study using the data collected by the cohort has been recently published, and seven others are currently underway:

- The interpersonal process model of intimacy, burden and communal motivation to care in a multinational group of informal caregivers [77]

- Predicting willingness to care and caregiver outcomes using an integrative theoretical framework of personality dispositions and environmental contextual factors [submitted for publication]

- Cross-country variations in caregiver values, meaning in life, illness beliefs, motivation and willingness to provide care & caregiver outcomes

- The influence of personal values, meaning in life, and illness beliefs on caregiver motivations, willingness & outcomes

- Does willingness to care fluctuate over time? A weekly diary study among informal caregivers

- The associations of dyadic coping strategies with caregiver's willingness to care and burden: a weekly diary study

- Health and labour market effects of informal care provision: a cross-country analysis

- Home care workers and the changing roles of informal caregivers: findings from the ENTWINE iCohort Survey on informal care

## Strengths and limitations

Having enrolled 1,872 caregivers and 402 care recipients, the ENTWINE iCohort is one of the largest multinational cohort studies of informal care yet conducted. One strength of this study is that it was designed and implemented by researchers across a range of social and behavioural science disciplines with expertise in informal care. Perhaps the main strength of this study lies in its examination of a wide range of personal, interpersonal, psychological, social, economic, and geographic factors, both at baseline and follow-up. This allows the assessment of interrelationships between these variables cross-sectionally and longitudinally while controlling for many potential confounders, and thus we can address a broad spectrum of research questions

related to informal care. Another strength of this study resides in its diverse sample of caregivers and care recipients from nine countries with differing long-term care systems and cultural norms and beliefs around informal care. Having such a diverse sample enhances the generalisability of the results whilst also enabling cross-country comparisons that can highlight both similarities and differences.

The present study shares the same limitations as other web-based cohort studies: low response rate, coverage bias, and non-response bias [78]. Despite the growing popularity of web-based surveys, debate continues about what can be considered an acceptable response rate [79,80]. Furthermore, web-based surveys generally achieve lower response rates than conventional surveys [81–83]. The response rates achieved in the two baseline surveys were above 40%. Although lower than desired, these rates still exceeded the average for web-based surveys, as reported in previous studies [84]. Several strategies were adopted to improve the response rate in the present study, including automatic login, sending two email reminders, and using simple web designs [85]. Nevertheless, one factor that might have affected the response rates was the survey fatigue associated with the proliferation of (web-based) surveys that ensued at the time of the study due to the COVID-19 pandemic [86]. All things considered, we believe that the response rates achieved in the baseline surveys were satisfactory.

One of the main limitations of web-based surveys is the inherent coverage bias [78]. Web-based survey studies necessitate Internet access, and therefore those with no access to the Internet do not have the opportunity to participate in such studies. Clearly, the extent of the coverage bias in web-based studies is proportional to the share of the target population without Internet access [87]. A suite of studies from the past decade shows that the coverage bias of web-based surveys has diminished in Europe as Internet access has become more widespread [88,89]. Now, Europe is one of the regions with the highest Internet penetration rates in the world. This is also true across all the countries studied, with Internet penetration rates ranging between 76%–94% [90]. Therefore, it was assumed at the design stage that the participating countries have sufficiently high Internet penetration and are well suited for web-based studies.

Another important concern with web-based surveys is non-response bias [91]. The risk of non-response bias in this study has likely been reduced by our efforts to improve the response rate. However, non-response bias in web-based surveys is not only a function of response rate but also of systematic differences between responders and non-responders [92]. Since the characteristics of non-respondents were unknown in this study, non-response bias could neither be estimated nor ruled out.

By design, and due to time and budget constraints, representative samples of caregivers and care recipients were not sought. However, best efforts were made to recruit diverse samples in order to capture the caregiving experiences of participants with different characteristics and care situations. Despite these efforts, females were overrepresented in both the caregiver and care recipient samples. However, this overrepresentation of females is consistently observed in studies on informal care [93–95] and reflects their actual dominance in informal care provision according to European statistics [13,96] and their greater willingness to participate in surveys in general [97,98].

Another limitation of the present study is the smaller-than-expected sample size in some countries, which places limits on the types of analyses that could be performed in some cases. However, approaches such as pooling data (e.g., across countries) are being considered to provide sufficient statistical power to perform some analyses (e.g., subgroup analysis).

The timing of baseline data collection in the middle of the COVID-19 pandemic is both a strength and a limitation. Some participants might have experienced significant changes to their caregiving and care-receiving experiences due to COVID-19 restrictive measures. Several studies to date have shown how COVID-19 exacerbated caregivers' burdens and strains and

negatively impacted their health, psychosocial, and financial outcomes [99,100]. Therefore, the COVID-19 situation and the countermeasures in place in each country at the time of data collection should be considered during data analysis and interpretation of findings. Despite the challenges, the timing of the study has also presented interesting opportunities to shed light on how caregivers' and care recipients' experiences have changed during a period of fluctuating infection rates and COVID-19 restrictions.

Finally, establishing and maintaining a longitudinal cohort of this size is costly, time-consuming, and resource-intensive. It was therefore difficult to follow the ENTWINE cohort beyond the 6-month follow-up period, thereby limiting the ability to examine longer-term patterns and outcomes within what is typically a dynamic caregiving experience.

## Conclusion

The ENTWINE iCohort Study is a multinational longitudinal cohort study investigating the multidimensional experiences of informal caregivers and care recipients in nine countries with divergent care systems and cultural orientations toward caregiving. This cohort provides a fertile opportunity to examine a wide range of personal, interpersonal, psychological, social, economic, and geographic factors and how they influence caregiving outcomes, including caregiver perceived gains and burdens, well-being and quality of life. We believe that the data collected in this study hold great promise for advancing caregiving research internationally and will inform the development of caregiver interventions and services, as well as policies aiming to improve caregiving experiences and outcomes.

## Supporting information

**S1 Table. A complete list of all caregiver organisations and advocacy groups involved in the recruitment of participants.**
(DOCX)

**S2 Table. Care recipient condition(s) at baseline as reported by their caregiver.**
(DOCX)

**S3 Table. Care recipient condition(s) at baseline.**
(DOCX)

## Acknowledgments

We would like to thank all study participants for taking the time to share their care experiences with us. We gratefully acknowledge all organisations that helped with the recruitment of participants by distributing the study surveys among its members. This includes organisations from: Germany (Allianz pflegende Angehörige), Italy (Anziani e non solo; Associazione C'ENTRO; Associazione de Banfield; Federazione Alzheimer Italia; Caffè Alzheimer Falconara; Associazione PAzienti LIberi dalle Neoplasie UROteliali [PALINURO]; Diabete Italia Onlus; Associazione Parkinson Marche; Rete Caregiver; Associazione Errante; ASPI Groane), Israel (Caregivers Israel; Camoni—Friends for Health), Ireland (Think Bodywhys; Family caregivers Ireland; Shine; Saint Vincent de Paul Ireland; Brain Tumour Ireland; Care Alliance Ireland; Chronic Pain Ireland; Muscular Dystrophy Ireland; Polio Survivors Ireland; Inclusion Ireland), Poland (Fundacja Hospicyjna; Stowarzyszenie Pomocy Psychologicznej Syntonia; Fundacja SM–walcz o siebie; Polskie Stowarzyszenie Diabetyków; Polskie Stowarzyszenie Pomocy Osobom z Chorobą Alzheimera; Fundacja W Związku Z Rakiem; Stowarzyszenie UNICORN; Fundacja ORCHidea; Niepelnosprawni.pl; Fundacja Instytut Praw Pacjenta i

Edukacji Zdrowotnej; Punkt Wsparcia Seniora; Centrum Wsparcia–opiekunów nieformal-nych i faktycznych), the Netherlands (MantelzorgNL; Zorgbelang; Vilans; Movisie), Sweden (Anhörigas riksförbund; Hjärt-Lungfonden; Diabetesfonden; Prostatacancerförbundet; Sar-komföreningen; Alzheimerfonden; Demenscentrum; ParkinsonFonden; AbbVie Sverigem; ParkinsonFörbundet; Anhörigstöd i Stockholms Län; Parkinson Skåne; Föreningen Balans Stockholm; Svenska Ödemförbundet; Mustaschkampen; Alzheimerfonden; Demensförbundet; Anhörigas Riksförbund; Svenska Diabetesförbundet; Riksförbundet Balans), and the United Kingdom (Carers Wales; Carers Trust; Carers Officers Learning Improvement Network for Wales; Shared Lives; Carers Outreach Service; North East Wales Carers Information Service [NEWCIS]; Mencap Cymru; Age Cymru; Gwynedd Council; Centre for Ageing and Dementia Research; North Wales Social Care and Wellbeing Services Improvement Collaborative; Shared Care Scotland; Centre for International Research on Care, Labour and Equalities [CIR-CLE]; Carers Federation; Join Dementia Research; Headway—The Brain Injury Association; The Brain Charity; MS Society UK; Breast Cancer UK). We also thank the entire ENTWINE consortium for their valuable comments and support.

## Author Contributions

**Conceptualization:** Saif Elayan, Eva Bei, Giulia Ferraris, Oliver Fisher, Mikołaj Zarzycki, Viola Angelini, Lena Ansmann, Erik Buskens, Mariët Hagedoorn, Milena von Kutzleben, Giovanni Lamura, Anne Looijmans, Robbert Sanderman, Noa Vilchinsky, Val Morrison.

**Data curation:** Erik Buskens, Mariët Hagedoorn, Giovanni Lamura, Robbert Sanderman, Noa Vilchinsky, Val Morrison.

**Formal analysis:** Saif Elayan.

**Funding acquisition:** Erik Buskens, Mariët Hagedoorn, Giovanni Lamura, Robbert Sander-man, Noa Vilchinsky, Val Morrison.

**Investigation:** Saif Elayan, Eva Bei, Giulia Ferraris, Oliver Fisher, Mikołaj Zarzycki, Viola Angelini, Lena Ansmann, Erik Buskens, Mariët Hagedoorn, Milena von Kutzleben, Gio-vanni Lamura, Robbert Sanderman, Noa Vilchinsky, Val Morrison.

**Methodology:** Saif Elayan, Eva Bei, Giulia Ferraris, Oliver Fisher, Mikołaj Zarzycki, Viola Angelini, Erik Buskens, Mariët Hagedoorn, Giovanni Lamura, Robbert Sanderman, Noa Vilchinsky, Val Morrison.

**Project administration:** Erik Buskens, Mariët Hagedoorn, Giovanni Lamura, Anne Looij-mans, Noa Vilchinsky, Val Morrison.

**Resources:** Erik Buskens, Mariët Hagedoorn, Giovanni Lamura, Anne Looijmans, Noa Vil-chinsky, Val Morrison.

**Software:** Saif Elayan.

**Supervision:** Viola Angelini, Erik Buskens, Mariët Hagedoorn, Giovanni Lamura, Noa Vil-chinsky, Val Morrison.

**Validation:** Saif Elayan, Eva Bei, Giulia Ferraris, Oliver Fisher, Mikołaj Zarzycki, Viola Angel-ini, Lena Ansmann, Erik Buskens, Mariët Hagedoorn, Milena von Kutzleben, Giovanni Lamura, Robbert Sanderman, Noa Vilchinsky, Val Morrison.

**Visualization:** Saif Elayan.

**Writing – original draft:** Saif Elayan.

**Writing – review & editing:** Saif Elayan, Eva Bei, Giulia Ferraris, Oliver Fisher, Mikołaj Zarzycki, Viola Angelini, Lena Ansmann, Erik Buskens, Mariët Hagedoorn, Milena von Kutzleben, Giovanni Lamura, Anne Looijmans, Robbert Sanderman, Noa Vilchinsky, Val Morrison.

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
