## [Decision Letter · Decision Letter 0]

26 Jun 2023

PONE-D-22-14610Cohort profile: The ENTWINE iCohort Study, a multinational longitudinal web-based study of informal carePLOS ONE

Dear Dr. Elayan,

Thank you for submitting your manuscript to PLOS ONE. After careful consideration, we feel that it has merit but does not fully meet PLOS ONE’s publication criteria as it currently stands. Therefore, we invite you to submit a revised version of the manuscript that addresses the points raised during the review process.

 First of all, I sincerely apologize for the delay in my response. Unfortunately, it was initially difficult to secure reviewers. Furthermore, several candidates who had initially accepted the task later stepped down, without delivering their reports. Please find the attached reports. Overall, both reviewers have provided positive evaluations of your contribution. However, they have also offered suggestions and requested clarifications that can help improve the paper. I kindly ask you to thoroughly read both reports and address the comments mentioned therein. If you are willing to do this, we will be ready to reconsider the paper for publication.

We look forward to receiving your revised manuscript.

Kind regards,

Matteo Lippi Bruni, PhD

Academic Editor

PLOS ONE

2. During our internal checks, the in-house editorial staff noted that you conducted research or obtained samples in another country. Please check the relevant national regulations and laws applying to foreign researchers and state whether you obtained the required permits and approvals. Please also state whether you approached an institutional review board (ethics committee) in Poland, Ireland and Greece before the study began.

Please address this in your ethics statement in both the manuscript and submission information. In addition, please ensure that you have suitably acknowledged the contributions of any local collaborators involved in this work in your authorship list and/or Acknowledgements. Authorship criteria is based on the International Committee of Medical Journal Editors (ICMJE) Uniform Requirements for Manuscripts Submitted to Biomedical Journals - for further information please see here: https://journals.plos.org/plosone/s/authorship.

3. Please amend your current ethics statement to address the following concerns:

a) Did participants provide their written or verbal informed consent to participate in this study?

6. Ethics statement appears in the Methods section of the manuscript AND at the end of the manuscript:

Your ethics statement should only appear in the Methods section of your manuscript. If your ethics statement is written in any section besides the Methods, please delete it from any other section.

Reviewers' comments:

Reviewer's Responses to Questions

**Comments to the Author**

1. Is the manuscript technically sound, and do the data support the conclusions?

Reviewer #1: Partly

Reviewer #2: Yes

2. Has the statistical analysis been performed appropriately and rigorously? 

Reviewer #1: Yes

Reviewer #2: N/A

3. Have the authors made all data underlying the findings in their manuscript fully available?

Reviewer #1: Yes

Reviewer #2: No

4. Is the manuscript presented in an intelligible fashion and written in standard English?

Reviewer #1: Yes

Reviewer #2: Yes

5. Review Comments to the Author

Reviewer #1: Summary

The paper presents the baseline cohort of the “ENTWINE iCohort Study” and accurately describes its design, recruitment methods, data collection procedures, measures, and early baseline statistics. The survey was conducted in nine European countries, namely Germany, Greece, Ireland, Israel, Italy, the Netherlands, Poland, Sweden, and the United Kingdom. The study comprised a web-based longitudinal survey (baseline + 6-month follow-up) and optional weekly diary assessments, conducted separately with caregivers and care recipients. Moreover, authors show a set of descriptive statistics of the respondents, focusing on the sample of caregivers and care-recipients separately.

General Comments

I think that the information collected in the survey may have potential for future research in the field. For instance, modules such as (i) cultural and psychosocial aspects and (ii) interpersonal processes are really interesting and not usually included in the existing international surveys on this topic. Moreover, information on perceived gains and caregiving burden may be useful to “measure” the pressure to which caregivers are exposed in doing their activities.

Even though I see potential for this work, there are some points which need a more in-depth discussion. In what follows I shall try to summarize my main comments.

1. The manuscript was classified as a “Research article”, but it describes in detail a new dataset, providing to the reader some descriptive statistics on the baseline data without answering any research questions. Honestly, I think it should be classified more properly as a presentation of a new database.

2. Authors write at p. 33:

“By design, and due to time and budget constraints, representative samples of caregivers and care recipients were not sought. However, best efforts were made to recruit diverse samples in order to capture the caregiving experiences of participants with different characteristics and care situations. Despite these efforts, females were overrepresented in both the caregiver and care recipient samples. However, this overrepresentation of females is consistently observed in studies on informal care [92–94] and reflects their actual dominance in informal care provision according to European statistics [13,95] and their greater willingness to participate in surveys in general [96,97]. The lack of representativeness in our samples can compromise the estimation of descriptive population parameters (i.e., population means and proportions) [98]. However, the estimation of association patterns, which is the main goal of the present study, is relatively robust to the bias introduced by the lack of representativeness [99]. In fact, it has been reported that with appropriate controlling for confounding only an extreme case of this bias can obscure association patterns [99] and apart from this particular case, a representative sample might not be needed [100] or desired [101]”

Personally, I think that having a representative sample is quite relevant in general. However, I recognize that some literature (cited in the paper) supports the thesis of the non-representativeness of the sample in specific cases.

Turning to what the authors claim above, it is not clear to me in particular this part:

“The lack of representativeness in our samples can compromise the estimation of descriptive population parameters (i.e., population means and proportions) [98]. However, the estimation of association patterns, which is the main goal of the present study, is relatively robust to the bias introduced by the lack of representativeness”

I would kindly ask the authors to explain this part more clearly. I do not understand what they refer to when they mention “association patterns”. In the paper, I see only descriptive statistics, for which the lack of representativeness may be an issue…

3. As mentioned by the authors, the information on caregiving was collected in a very “special” period, during the first waves of the Covid-19 pandemics. They discuss in the text strengths and limitations related to the timing of the survey, claiming that some groups of participants might have experienced significant changes to their caregiving and care receiving experiences due to COVID-19 restrictive measures. Moreover, caregivers and in particular care recipients are among the most vulnerable groups in the population, and were more exposed to the negative consequences of the pandemics. Due to the peculiarity of the period, it should be very helpful to have further rounds of data that allow to look at the long-term patterns of informal caregiving.

4. I would include in the paper some indications concerning the availability of the dataset. Is it publicly accessible? Where? As highlighted in the “Submission Guidelines” of the Journal, “the database must be open-access and hosted somewhere publicly accessible”.

Minor:

• p. 6 line 111: “MH, VM, RS, and NV”: These acronyms are not explained before in the text. Please clarify the meaning.

Reviewer #2: The paper describes the design, recruitment methods, data collection procedures, measures, and early baseline findings of the cohort study. The authors adopted ENTWINE icohort Study. The analysis involves 9 countries and studies the period from 14 August 2020 to 31 August 2021.

In the complex, the paper is well-written and well-finished in each section, I just recommend some modifications or further clarification that could ameliorate the work.

Line 125. I suggest the authors to better explain if August 2020-December 2021 is the baseline period only or if it included also the 6 months of follow-up. Anyway, it resulted in conflict with the study period indicated in the abstract. Or just refer to the dedicated section.

Line 242. It is not clear when the recruitment phase ended: that is, is it staggered recruitment? Are there people who had completed the baseline survey while others still have to be recruited?

Line 249. How are classified people who choose "none of the above", were drop from the sample?

Section “Baseline cohort characteristics”.

-I suggest adding information about some indices included in the analyses, such as "WHO-5 score" and "Katz ADL index", in particular, if they were somehow calculated or retrieved somewhere (in the survey or elsewhere).

-It is not clear if follow-up phases are ended too. Consequently, aren't the results of follow-up surveys and (optional) weekly diary assessments available yet? Are they not presented in the results paragraph?

6. PLOS authors have the option to publish the peer review history of their article (what does this mean?). If published, this will include your full peer review and any attached files.

Reviewer #1: No

Reviewer #2: No

---

## [Author Response · Author response to Decision Letter 0]

25 Aug 2023

Journal Requirements

Comment 1: Please ensure that your manuscript meets PLOS ONE's style requirements, including those for file naming. The PLOS ONE style templates can be found at: https://journals.plos.org/plosone/s/file?id=wjVg/PLOSOne_formatting_sample_main_body.pdf and https://journals.plos.org/plosone/s/file?id=ba62/PLOSOne_formatting_sample_title_authors_affiliations.pdf.

Response: We have carefully revised our manuscript to ensure compliance with PLOS ONE’s style requirements and made the necessary adjustments throughout the manuscript.

Comments 2 and 3: 

• During our internal checks, the in-house editorial staff noted that you conducted research or obtained samples in another country. Please check the relevant national regulations and laws applying to foreign researchers and state whether you obtained the required permits and approvals. Please also state whether you approached an institutional review board (ethics committee) in Poland, Ireland and Greece before the study began.

Please address this in your ethics statement in both the manuscript and submission information. In addition, please ensure that you have suitably acknowledged the contributions of any local collaborators involved in this work in your authorship list and/or Acknowledgements. Authorship criteria is based on the International Committee of Medical Journal Editors (ICMJE) Uniform Requirements for Manuscripts Submitted to Biomedical Journals - for further information please see here: https://journals.plos.org/plosone/s/authorship.

• Please amend your current ethics statement to address the following concerns: a) Did participants provide their written or verbal informed consent to participate in this study? b) If consent was verbal, please explain i) why written consent was not obtained, ii) how you documented participant consent, and iii) whether the ethics committees/IRB approved this consent procedure.

Response:

In regard to the concern about authorship and acknowledgement of contributions from local collaborators, we wish to affirm that every author included in our manuscript fulfils the authorship criteria set forth by the International Committee of Medical Journal Editors (ICMJE). Each listed author has contributed significantly to the study, participated actively in drafting or revising the article, given approval for the version to be published, and agreed to be accountable for all aspects of the work. Additionally, all local collaborators who have contributed to the project, but did not meet the full criteria for authorship as per the ICMJE, have been duly recognised in the Acknowledgements section of the manuscript.

Turning to the matter of the consent procedure, and the ethical approvals in Poland, Ireland, and Greece, we would like to confirm that all required permits and approvals have been duly obtained in every country where the research was conducted. We have made necessary modifications to our ethics statement within the manuscript and submission information to illustrate this more explicitly.

The revised ethics statement (pages 21¬–22, lines 363–383) now reads (changes highlighted in yellow):

“Ethical approvals for the study and its consent procedure were obtained from multiple institutions across the participating countries: Institutional Review Board, Bangor University, The UK; NHS Research Ethics and Governance Committee, The UK (reference number: 20/WA/0006); Central Ethics Review Board non-WMO studies, University Medical Center Groningen, The Netherlands (reference number: 201900810); Bar-Ilan University, Faculty of Social Sciences, Department of Psychology, Ethics Committee, Israel (reference number: 36-20); Commissione Etica per la Ricerca in Psicologia (CERPS), Università Cattolica del Sacro Cuore di Milano, Italy (reference number: 31-20); Swedish Ethical Review Authority, Uppsala University, Sweden (reference number: 2020-04569); and Medical Ethics Committee, University of Oldenburg, Germany (reference number: 2020-155). In Poland, the ethical board of the Institute of Psychology at the University of Wrocław recognised the UK NHS Research Ethics Approval as sufficient for conducting the study. Similarly, in Greece, the Department of Psychology at the University of Crete acknowledged the ethical approval granted by the University Medical Center Groningen on the basis of approvals from other European nations. In Ireland, Care Alliance Ireland, a registered charity, deemed the ethical approval from Bangor University sufficient for participant recruitment and study conduct. All these ethical approvals were obtained before initiating recruitment, enrolment, and data collection. All participants were required to give informed written consent via the survey platform Questback before they could gain access to the surveys. Individuals who did not provide consent were denied participation and redirected to a page thanking them for their interest.”

Comments 4 and 5: 

• In your Data Availability statement, you have not specified where the minimal data set underlying the results described in your manuscript can be found. PLOS defines a study's minimal data set as the underlying data used to reach the conclusions drawn in the manuscript and any additional data required to replicate the reported study findings in their entirety. All PLOS journals require that the minimal data set be made fully available. For more information about our data policy, please see http://journals.plos.org/plosone/s/data-availability.

• We note that you have indicated that data from this study are available upon request. PLOS only allows data to be available upon request if there are legal or ethical restrictions on sharing data publicly. For more information on unacceptable data access restrictions, please see http://journals.plos.org/plosone/s/data-availability#loc-unacceptable-data-access-restrictions.

Response:

The minimal data set required to replicate the findings reported in the article has been deposited and made publicly available without restriction on Zenodo, an open-access repository. The revised data availability statement is as follows:

“A de-identified minimal data set underlying the results described in this article is available on the Zenodo repository at https://doi.org/10.5281/zenodo.8170318. Additional baseline data requests can be made after September 2024 to the ENTWINE data access committee via email at entwine@umcg.nl.”

We believe that the revised Data Availability statement fully complies with PLOS’ data policy.

Comment 6:

Ethics statement appears in the Methods section of the manuscript AND at the end of the manuscript: Your ethics statement should only appear in the Methods section of your manuscript. If your ethics statement is written in any section besides the Methods, please delete it from any other section.

Response: We have revised the manuscript to ensure that the ethics statement appears solely in the Methods section in accordance with the journal requirements. 

Comment 7: 

Response: The reference list has been reviewed, and necessary adjustments have been implemented. We confirm that the list is now complete and correct. A detailed account of the changes made can be found in the “Reference list revisions” section at the end of this letter.

Response to Reviewer #1 comments:

Comment 1: The manuscript was classified as a “Research article”, but it describes in detail a new dataset, providing to the reader some descriptive statistics on the baseline data without answering any research questions. Honestly, I think it should be classified more properly as a presentation of a new database.

Response:

We appreciate the reviewer’s point. In classifying our manuscript, we chose “Research Article” because it seemed the most suitable among the options provided by the Journal’s submission system, which includes Clinical Trial, Research Article, Collection Review & Overview, Registered Report Protocol, Registered Report, Lab Protocol, and Study Protocol. We also looked at similar articles profiling cohort studies published in PLoS ONE. All the ones we found were categorised as “Research Article”. Therefore, we decided to follow this practice.

Comment 2: Authors write at p. 33: “By design, and due to time and budget constraints, representative samples of caregivers and care recipients were not sought. However, best efforts were made to recruit diverse samples in order to capture the caregiving experiences of participants with different characteristics and care situations. Despite these efforts, females were overrepresented in both the caregiver and care recipient samples. However, this overrepresentation of females is consistently observed in studies on informal care [92–94] and reflects their actual dominance in informal care provision according to European statistics [13,95] and their greater willingness to participate in surveys in general [96,97]. The lack of representativeness in our samples can compromise the estimation of descriptive population parameters (i.e., population means and proportions) [98]. However, the estimation of association patterns, which is the main goal of the present study, is relatively robust to the bias introduced by the lack of representativeness [99]. In fact, it has been reported that with appropriate controlling for confounding only an extreme case of this bias can obscure association patterns [99] and apart from this particular case, a representative sample might not be needed [100] or desired [101]”

Personally, I think that having a representative sample is quite relevant in general. However, I recognize that some literature (cited in the paper) supports the thesis of the non-representativeness of the sample in specific cases.

Turning to what the authors claim above, it is not clear to me in particular this part:

“The lack of representativeness in our samples can compromise the estimation of descriptive population parameters (i.e., population means and proportions) [98]. However, the estimation of association patterns, which is the main goal of the present study, is relatively robust to the bias introduced by the lack of representativeness”

I would kindly ask the authors to explain this part more clearly. I do not understand what they refer to when they mention “association patterns”. In the paper, I see only descriptive statistics, for which the lack of representativeness may be an issue…

Response: 

We appreciate the reviewer’s insightful comment and the request for further clarification. Indeed, our manuscript primarily focuses on descriptive statistics. However, the purpose of including these statistics is not to estimate population parameters but rather to provide an in-depth description of the baseline cohort in the ENTWINE iCohort Study. The descriptive statistics presented in this paper are intended to depict the characteristics of the baseline cohort in the study and are not intended to make claims about the broader population of caregivers in the countries studied.

The reference to association patterns and the influence of sample representativeness on their estimation in the manuscript is relevant to the overarching goal of the broader ENTWINE iCohort Study rather than the specific objectives of this paper. As stated in the manuscript (page 6, lines 112–117), the ENTWINE iCohort Study seeks to examine the associations among various variables in our study (e.g., personal, interpersonal, psychological, social, economic, and geographic) and caregiving experiences and outcomes. 

We recognise the confusion caused by our wording and appreciate the reviewer bringing it to our attention. Upon reconsideration, we concur that the specific part of the paragraph that the reviewer highlighted could indeed introduce unnecessary complexity, leading to confusion regarding our paper’s primary aim. Therefore, we have decided to omit this particular part from the manuscript. We believe this decision will enhance the clarity of our research objective and dispel any potential confusion. We are grateful for the reviewer’s comment, which led us to this conclusion.

Comment 3: As mentioned by the authors, the information on caregiving was collected in a very “special” period, during the first waves of the Covid-19 pandemics. They discuss in the text strengths and limitations related to the timing of the survey, claiming that some groups of participants might have experienced significant changes to their caregiving and care receiving experiences due to COVID-19 restrictive measures. Moreover, caregivers and in particular care recipients are among the most vulnerable groups in the population, and were more exposed to the negative consequences of the pandemics. Due to the peculiarity of the period, it should be very helpful to have further rounds of data that allow to look at the long-term patterns of informal caregiving.

Response: We agree with the reviewer’s point. The COVID-19 pandemic indeed presented peculiar circumstances that could affect caregiving situations and dynamics. We acknowledged this in our manuscript and discussed how these conditions might influence our results (pages 36¬–37, lines 590–599). 

As mentioned in the manuscript (page 37, lines 600–603), we fully recognise the importance of following the caregiving experience longitudinally to capture its dynamic nature. However, we also highlighted the challenges in maintaining such a longitudinal cohort study, including the substantial costs, time commitment, and resource intensity. Due to these factors, it was difficult to follow the ENTWINE cohort beyond the 6-month follow-up period.

While we share the reviewer’s interest in observing long-term patterns in caregiving, the realities of our study design and the resources available have imposed some limitations on our ability to conduct further data collection at this time.

Comment 4: I would include in the paper some indications concerning the availability of the dataset. Is it publicly accessible? Where? As highlighted in the “Submission Guidelines” of the Journal, “the database must be open-access and hosted somewhere publicly accessible”.

Response: 

We appreciate the reviewer’s concern about data accessibility. As addressed in response to a previous comment in this letter, the data set underpinning this article has been made publicly accessible. It has been deposited on Zenodo, an open-access repository. The revised Data Availability Statement is now as follows: 

“A de-identified minimal data set underlying the results described in this article is available on the Zenodo repository at https://doi.org/10.5281/zenodo.8170318. Additional baseline data requests can be made after September 2024 to the ENTWINE data access committee via email at entwine@umcg.nl.”

With this amendment to the data availability statement, we believe we are in full compliance with the Journal’s submission guidelines concerning data availability.

Comment 5: p. 6 line 111: “MH, VM, RS, and NV”: These acronyms are not explained before in the text. Please clarify the meaning.

We thank the reviewer for their comment regarding the usage of "MH, VM, RS, and NV" in our manuscript. In this context, these are the initials of some of the co-authors, serving to reference their previous collaborations. This was intended to provide additional context on the emergence of the consortium without compromising the brevity of the text. However, recognising that this could potentially confuse the reader, and since this information is not integral to understanding the research presented, we have decided to remove the sentence mentioning these initials from the manuscript.

Response to Reviewer #2 comments:

Comment 1: Line 125. I suggest the authors to better explain if August 2020-December 2021 is the baseline period only or if it included also the 6 months of follow-up. Anyway, it resulted in conflict with the study period indicated in the abstract. Or just refer to the dedicated section.

Response: We greatly appreciate the reviewer’s astute observation and the highlighting of potential confusion regarding the dates. The date range from August 2020 to December 2021, as referenced in the line noted by the reviewer, corresponds to the entire duration of the study, encompassing both the baseline data collection and the 6-month follow-up. Conversely, the date range specified in the abstract pertains solely to the baseline data collection period. 

We understand how this discrepancy might lead to ambiguity, and we have addressed it by making the distinction clearer. As such, we have revised the paragraph that contains the line highlighted by the reviewer to explicitly state that the date range from August 2020 to December 2021 relates to the entire data collection period. We believe that this amendment successfully resolves the issue raised by the reviewer. The revised paragraph (page 7, lines 131–142) now reads as follows (changes highlighted in yellow):

“The ENTWINE iCohort Study is a multinational web-based cohort study with an intensive longitudinal design that combines a two-wave panel survey (baseline + 6 months follow-up) with optional weekly diary assessments. The entire data collection period spanned from August 2020 to December 2021. The cohort includes caregivers and care recipients from nine countries, including those represented in the ENTWINE consortium (the United Kingdom, the Netherlands, Italy, Sweden, and Israel) and four other European countries (Germany, Greece, Poland, and Ireland). Participating countries were selected to represent different geographic areas (i.e., North, East, West, and South) and typologies of welfare states in Europe [40]. The initial plan was to administer the surveys in paper and web-based formats; however, the former format was suspended due to COVID-19 pandemic restrictions. The detailed protocol of the study has been published elsewhere [41].”

Comment 2: Line 242. It is not clear when the recruitment phase ended: that is, is it staggered recruitment? Are there people who had completed the baseline survey while others still have to be recruited?

Response: 

We thank the reviewer for their comment. Indeed, the recruitment for the baseline survey was staggered over the 12-month baseline data collection period. To provide clearer information in our manuscript, we revised the paragraph containing the line pointed out by the reviewer (pages 22¬–23, lines 386–393) to read as follows:

“Baseline recruitment was staggered over the 12-month baseline data collection period from 14 August 2020 to 31 August 2021. The start dates for recruitment varied by country as follows: the UK, Ireland, and Poland on 14 August 2020; Italy on 25 August 2020; the Netherlands on 14 October 2020; Sweden on 23 October 2020; Greece on 31 October 2020; and Germany and Israel on 16 February 2021. Baseline recruitment was concluded on 31 May 2021 in all countries except Germany and Israel. In these two countries, recruitment was extended until 31 August 2021, thereby ensuring a recruitment period of at least six months in all participating countries.”

Comment 3: Line 249. How are classified people who choose “none of the above”, were drop from the sample?

Response: 

We thank the reviewer for their comment. Individuals who selected the “None of the above” option during the eligibility screener were considered ineligible for inclusion in the study, as they did not identify themselves as either caregivers or care recipients. Therefore, these individuals were not dropped post-inclusion; rather, they were excluded from the outset based on the eligibility criteria. 

We acknowledge that our original manuscript could have more explicitly detailed our exclusion process. In response to the reviewer’s comment, we have incorporated an additional sentence after the line pointed out by the reviewer. The added sentence (page 13, lines 261-263) reads as follows:

“Conversely, respondents who selected “None of the above” were deemed ineligible, as they did not meet the criteria of being caregivers or care recipients, and thus, were screened out and not included in the study.” 

We trust this amendment dispels any ambiguity regarding our exclusion criteria and clarifies the process for the reader.

Comment 4: I suggest adding information about some indices included in the analyses, such as “WHO-5 score” and “Katz ADL index”, in particular, if they were somehow calculated or retrieved somewhere (in the survey or elsewhere).

Response 4: 

We highly value the constructive suggestion from the reviewer to provide more comprehensive information about the indices used in our analysis. In response to this recommendation, we have made the following enhancements to our manuscript:

We supplemented our Methods section with detailed information about the indices used in our analysis, their usage in our study, and how their scores were derived from our survey data. This additional information aims to ensure full transparency on the utilisation of these indices in our analysis. The text that we have added to the Methods section (pages 18–20, lines 294–327) reads as follows:

In this article, we focus on characterising the cohort at baseline in terms of key sociodemographics (i.e., age, gender, marital status, education and employment status), caregiving situation characteristics (i.e., care intensity, primary caregiving, and kinship type), and care recipient’s health condition(s) and dependency. Furthermore, we consider the willingness and ability to provide care, gains and burden from caregiving, and well-being.

The dependency level of care recipients was assessed using the Katz Index of Independence in Activities of Daily Living (ADL). This instrument measures independence in performing six basic ADL: bathing, dressing, toileting, transferring, continence, and feeding. Each of these ADL is scored as 1 for independence and 0 for dependence, and the Katz index score is obtained by totalling these individual scores [47]. The score indicates complete independence (score = 6), partial dependence (score 3–5), or severe dependence (score ≤ 2) [73]. 

Caregiver willingness and ability to provide care were measured using the Willingness to Care Scale. The scale comprises 30 items, each representing a specific emotional, instrumental, or nursing care task. The ability to perform each task was scored as either “able” or “not able”, and the willingness to carry out the task was rated on a 5-point Likert scale, ranging from “completely unwilling” (1) to “completely willing” (5). The ability-to-care score was computed by summing up the “able” responses, while the willingness-to-care score was calculated by averaging the ratings on the 5-point Likert scale [51].

Caregiver gains were assessed using the GAINS scale. The scale comprises ten items, each scored on a 4-point Likert scale ranging from 0 (“not at all”) to 3 (“a lot”). The total score was calculated by adding the points for each item [53]. Caregiver burden was measured using the Short-Form Zarit Burden Interview (ZBI-12). This instrument includes 12 items, each rated on a 6-point Likert scale from 0 (“never”) to 5 (“nearly always”). The total score was obtained by summing the ratings of each item, resulting in a possible range from 0 to 60, with higher scores indicating higher levels of burden [54].

The well-being of caregivers and care recipients was evaluated using the World Health Organisation-Five Well-Being Index (WHO-5). The instrument consists of five positively worded statements related to well-being, each scored on a scale of 0 (“at no time”) to 5 (“all of the time”). The sum of the scores for the five items (raw score) is multiplied by four, resulting in a percentage score ranging from 0 (worst imaginable well-being) to 100 (highest imaginable well-being) [52].

We also revised the footnotes in Tables 2 and 3 to include an expanded description of the Katz Activities of Daily Living (ADL) and WHO-5 Well-Being indices. These revisions elucidate the score range and interpretation of these indices, thereby furnishing the reader with the necessary information to comprehend the indices and interpret their results. 

The added sentences in the footnotes for Tables 2 and 3 are as follows:

“ADL: Activities of Daily Living (eating, bathing, dressing, toileting, transferring, and continence).” (page 28, lines 450–451) & (page 32, lines 506–507)

“The range of score is from 0 (worst imaginable well-being) to 100 (highest imaginable well-being).” (page 28, line 452) (page 32, line 508)

“WHO-5 score ≤ 50 indicates suboptimal well-being [76].” (page 28, line 453) & (page 32, line 509)

“The score ranges from 0 (dependence in all ADLs) to 6 (total independence).” (page 28, lines 454¬–455) & (page 32, line 510)

We trust that these amendments, in conjunction with the information and references already provided in Table 1, ensure that the reader is furnished with sufficient insight into the indices utilised in our research. We are confident that these revisions will notably enhance the clarity of our research findings and the transparency of our analysis.

Comment 5: It is not clear if follow-up phases are ended too. Consequently, aren’t the results of follow-up surveys and (optional) weekly diary assessments available yet? Are they not presented in the results paragraph?

Response:

We thank the reviewer for raising this point. We want to clarify that the follow-up phase of our study has indeed been completed. However, the current manuscript focuses solely on presenting the baseline results of the ENTWINE iCohort Study. The data gathered from the follow-up surveys and optional weekly diary assessments will be analysed and presented in forthcoming publications. We value the reviewer’s interest in our study and anticipate sharing these future findings.

In response to the reviewer’s first comment, we incorporated a sentence in the manuscript to state explicitly: “The entire data collection period spanned from August 2020 to December 2021” (Page 7, lines 133–134). By including this information, it should be evident to the reader that all data collection activities, including the follow-up surveys and optional weekly diary assessments, were concluded by December 2021.

Additional revisions from the authors (highlighted in blue in the revised manuscript).

We have made minor revisions to enhance the manuscript's consistency, clarity, and accuracy. The revisions are detailed below:

Formatting revisions:

ZIP or Postal Codes, street addresses, and building/office numbers have been removed from the author affiliations on the title page, in accordance with the PLOS ONE guidelines.

The notation of percentages in the revised manuscript has been standardized, with "per cent" being replaced by "%", when possible, for consistency.

The date format in the main body of the manuscript has been standardized to follow the "Day Month Year" style for uniformity.

Revisions to tables and figures:

A missing element in the name of the IOS scale in Table 1 has been rectified. The complete name is now presented as "Inclusion of Other in the Self Scale (IOS)"

A typo in the title of Fig 1 has been corrected to: "Fig 1. Number of participants enrolled in the cohort in each month of the baseline recruitment period". (page 23, lines 404–405)

In Tables 2 and 3, the row "WHO-5 score < 50" (now "Score ≤ 50" in the revised manuscript), representing the proportion of participants with a WHO-5 score below 50, has been nested under the "WHO-5 score" row, which reflects the mean WHO-5 score. This optimization removes redundancy, aids interpretation, and enhances the data presentation.

The revised sentences are as follows:

“Given the complex nature of the informal caregiving experience, it is crucial that any attempt to comprehend it takes into account the various intertwined factors that contribute to shaping this experience.” (page 6, lines 109–111)

“Organisations and groups were not offered any incentives for recruitment efforts.” (page 9, line 185)

“Those confirmed as eligible were then required to read a participant information sheet and a consent form and to provide their e-mail address.” (page 11, lines 211–212)

“Upon providing their consent and email address, each participant was sent an invitation email.” (page 11, lines 212–213)

Due to a typo, the year of the date mentioned on page 11, line 219, has been corrected to 2021.

“The core module assessed sociodemographics and aspects of the care situation and included validated questionnaires targeting key dimensions of the caregiving experience (e.g., well-being, willingness to provide care, and relationship characteristics).” (page 13, lines 271–273).

“The decision to randomly assign three out of the four additional modules was made to shorten the survey and reduce its response burden, thereby improving the response rate.” (page 14, lines 280–282)

“Out of the 6,260 respondents who completed the eligibility survey, 2,893 (46.2%) did not meet the study’s eligibility criteria. This left 3,367 participants available for the study, comprising 2,731 caregivers (81.1%) and 636 care recipients (18.9%).” (pages 23–24, lines 410–413).

In response to a reviewer's comment, additional text was incorporated into the Methods section. Consequently, redundant information on page 28, lines 458–461, was removed, and the acronyms “WHO-5” and “ZBI-12”, now defined in the added text, were employed.

The section formerly titled "Ongoing Projects" has been updated to account for changes since the initial submission over a year ago. The title of the section has been modified to "Published Studies and ongoing Projects". To accurately depict the current status of the projects, we have denoted one as published and revised the titles of several others. These modifications are marked in blue in the revised manuscript (page 33, lines 513¬¬¬¬–532).

“The response rates achieved in the two baseline surveys were above 40%. Although lower than desired, these rates still exceeded the average for web-based surveys, as reported in previous studies” (page 34, lines 551–553)

Reference list revisions 

We implemented the following changes to the reference list in our manuscript:

1. We updated the access date format for all online sources from (dd mm yyyy) to (yyyy mm dd) and replaced the term “Available” with “Available from”, following PLoS ONE citation style.

2. Deletion of four references: In response to a reviewer's feedback, a paragraph in our manuscript was removed. Accordingly, four references associated with this paragraph were also deleted. These references were as follows:

Cheung KL, ten Klooster PM, Smit C, de Vries H, Pieterse ME. The impact of non-response bias due to sampling in public health studies: A comparison of voluntary versus mandatory recruitment in a Dutch national survey on adolescent health. BMC Public Health. 2017;17: 276. doi:10.1186/s12889-017-4189-8

Heiervang E, Goodman R. Advantages and limitations of web-based surveys: evidence from a child mental health survey. Soc Psychiatry Psychiatr Epidemiol. 2011;46: 69–76. doi:10.1007/s00127-009-0171-9

Richiardi L, Pizzi C, Pearce N. Commentary: Representativeness is usually not necessary and often should be avoided. Int J Epidemiol. 2013;42: 1018–1022. doi:10.1093/ije/dyt103

Rothman KJ, Gallacher JE, Hatch EE. Why representativeness should be avoided. Int J Epidemiol. 2013;42: 1012–1014. doi:10.1093/ije/dys223

3. Addition of a new reference: We have incorporated the following reference pertaining to a published project that made use of our study data:

Reference 77 (Page 53, lines 911-914)

Ferraris G, Bei E, Coumoundouros C, Woodford J, Saita E, Sanderman R, et al. The interpersonal process model of intimacy, burden and communal motivation to care in a multinational group of informal caregivers. J Soc Pers Relat. 2023;0: 1–23. doi: 10.1177/02654075231174415

4. Rectification of some references: We have rectified several references in our list to ensure conformity with the PLoS ONE citation style. The details of these changes are as follows:

Reference 4 (Page 42, lines 687-689)

European Commission, Eurostat, Corselli-Nordblad L, Strandell H. Ageing Europe: looking at the lives of older people in the EU: 2020 edition. Publications Office; 2020. doi: 10.2785/628105

Reference 5 (Page 42, lines 690-691)

OECD. Life expectancy at 65 (indicator) [Internet]. 2022 [cited 20 Mar 2022]. Available from: https://doi.org/10.1787/0e9a3f00-en

Reference 7 (Page 43, lines 696-701)

Geerts J, Willemé P, Pickard L, King D, Comas-Herrera A, Wittwer J, et al. Projections of Use and Supply of Long-Term Care in Europe: Policy Implications [Internet]. Brussels: Centre for European Policy Studies; 2012 [cited 20 Mar 2022]. 14 p. ENEPRI Policy Brief No.: 12. Available from: https://www.ceps.eu/ceps-publications/projections-use-and-supply-long-term-care-europe-policy-implications

Reference 8 (Page 43, lines 702-706)

Wittenberg R, Pickard L, Malley J, King D, Comas-Herrera A, Darton R. Future Demand for Social Care, 2005 to 2041: Projections of Demand for Social Care for Older People in England [Internet]. Personal Social Services Research Unit; 2008 [cited 20 Mar 2022] 9 p. Discussion Paper No.: 2514. Available from: https://kar.kent.ac.uk/88828/1/dp2514.pdf

Reference 13 (Page 44, lines 720-722)

European Commission, Directorate-General for Employment, Social Affairs and Inclusion, Zigante V. Informal care in Europe: exploring formalisation, availability and quality. Publications Office; 2018. doi: 10.2767/78836

Reference 18 (Page 44, lines 734-736)

Andersen R, Newman JF. Societal and Individual Determinants of Medical Care Utilization in the United States. Milbank Q. 2005;83. doi: 10.1111/j.1468-0009.2005.00428.x

Reference 35 (Page 47, lines 787-790)

Rodrigues R, Schulmann K, Schmidt A, Kalavrezou N, Matsaganis M. The indirect costs of long-term care [Internet]. European Commission; 2013 [cited 20 Mar 2022] 42 p. Research Note No.: 8/2013. Available from: https://www.euro.centre.org/publications/detail/415

Reference 42 (Page 48, lines 812-815)

Eurostat. Population on 1 January by age and sex [Internet]. 2022 [cited 2022 Mar 1]. Available from: https://ec.europa.eu/eurostat/databrowser/bookmark/97ede977-c619-4d31-bc7e-eb898695703b?lang=en

Reference 90 (Page 55, lines 947-949)

World Bank. Individuals using the Internet (% of population) [Internet]. 2022 [cited 2022 Mar 4]. Available from: https://data.worldbank.org/indicator/IT.NET.USER.ZS

Reference 98 (Page 56, lines 972-974)

Johansson S, Gulliksen J, Gustavsson C. Survey methods that enhance participation among people with disabilities. diva2:1362513 [Preprint]. 2019 [cited 2022 Mar 21]. Available from: https://urn.kb.se/resolve?urn=urn:nbn:se:kth:diva-262818

---

## [Decision Letter · Decision Letter 1]

26 Oct 2023

Cohort profile: The ENTWINE iCohort Study, a multinational longitudinal web-based study of informal care

PONE-D-22-14610R1

Dear Dr. Elayan,

We’re pleased to inform you that your manuscript has been judged scientifically suitable for publication and will be formally accepted for publication once it meets all outstanding technical requirements.

Kind regards,

Matteo Lippi Bruni, PhD

Academic Editor

PLOS ONE

Additional Editor Comments (optional):

Reviewers' comments:

Reviewer's Responses to Questions

**Comments to the Author**

1. If the authors have adequately addressed your comments raised in a previous round of review and you feel that this manuscript is now acceptable for publication, you may indicate that here to bypass the “Comments to the Author” section, enter your conflict of interest statement in the “Confidential to Editor” section, and submit your "Accept" recommendation.

Reviewer #1: All comments have been addressed

Reviewer #2: All comments have been addressed

2. Is the manuscript technically sound, and do the data support the conclusions?

Reviewer #1: Yes

Reviewer #2: Partly

3. Has the statistical analysis been performed appropriately and rigorously? 

Reviewer #1: Yes

Reviewer #2: Yes

4. Have the authors made all data underlying the findings in their manuscript fully available?

Reviewer #1: Yes

Reviewer #2: Yes

5. Is the manuscript presented in an intelligible fashion and written in standard English?

Reviewer #1: Yes

Reviewer #2: Yes

6. Review Comments to the Author

Reviewer #1: I thank the authors for the serious work incorporating my suggestions into the paper. I enjoyed reading the revised version.

Other than that, I think the paper is ready for publication.

Reviewer #2: The authors revised the manuscript "Cohort profile: The ENTWINE iCohort Study, a multinational longitudinal web-based study of informal care" on the base of reviewers' comments. The new version better meets the indications and suggestions.

Concerning the previous comments and recommendations, I consider that the authors made a work on the paper that returns a clearer and more readable analysis.

In regards to responses to comments 1-2-5, in the new version submitted all issues related to periods (base period, recruitment period, and follow-up) are been cleared and better explained. And also all related questions about staggered recruitment and the content of the results discussion.

The added sentence about the "None of the above" (response to comment 3) allows the reader to identify and understand the category.

The additional information about indices employed in the analysis supports in the reading and understanding.

I appreciate the extension of the section related to information about the Katz index and WHO-5 (and integration in footnotes).

As concerned with the work in the complex and the structure of the manuscript, I agree with the issue raised by reviewer #1 concerning the classification of the manuscript as a “Research article”. I consider that, since it mainly consists of a detailed description of a new dataset, it should be identified and reclassified as a presentation of a new database.

7. PLOS authors have the option to publish the peer review history of their article (what does this mean?). If published, this will include your full peer review and any attached files.

Reviewer #1: No

Reviewer #2: No

---

## [Editor Report · Acceptance letter]

9 Jan 2024

PONE-D-22-14610R1 

PLOS ONE

Dear Dr. Elayan, 

I'm pleased to inform you that your manuscript has been deemed suitable for publication in PLOS ONE. Congratulations! Your manuscript is now being handed over to our production team.

Kind regards, 

on behalf of

Dr. Matteo Lippi Bruni 

Academic Editor

PLOS ONE